

# Simulation of the heat mitigation potential of unsealing measures in cities by parameterizing grass grid pavers for urban microclimate modelling with ENVI-met (V5)

Nils Eingrüber[1], Alina Sophie Domm[1], Wolfgang Korres[1], Karl Schneider[1]

[1]University of Cologne, Institute of Geography, Hydrogeography and climatology research group, 50923, Cologne, Germany

*Correspondence to*: Nils Eingrüber (*nils.eingrueber@uni-koeln.de*)

**Abstract.** Many urban areas are characterized by both a growing population and an intensification of summer heat events in the context of climate change. Thus, more and more people are exposed to heat stress and corresponding health consequences. Measures for climate change adaptation such as unsealing strategies are needed in the existing urban fabric to reduce sensible heat flux by increasing latent heat flux to cool down the urban environment without requiring additional space or changing the basic function of the area. Unsealing measures like grass grid pavers (GGPs) can also help to reduce flooding risks due to increased infiltration and water storage capacities. Up to now, a parameterization of GGPs for microclimatic simulations is not available. To fill this research gap, we here present a new GGP model parameterization developed for the fluid dynamics microclimate ENVI-met model based on field measurements with double-ring infiltrometers etc. To analyse the microclimatic effects and the cooling potential of this GGP parameterization, scenario analyses were performed using a validated ENVI-met model setup for an urban high-density study area in Cologne/Germany. An extreme scenario was designed to address the maximum cooling potential of the GGPs in comparison to the dominant sealed asphalt surfaces in the study area, and a more realistic scenario with a usage-compatible installation of GGPs in the model domain only in side streets and inner courtyards while main streets remain sealed We found a maximum cooling potential of up to -20.1 K for ground surface temperature and up to -7.1 K for air temperature in 1 m above ground level for a simulated 3-day heat wave in summer 2022 which represents a 20-year heat event in Cologne. On spatial average, a decrease of up to -11.1 K for surface temperature and up to -2.9 K for air temperature was determined. On temporal average for the 3-day heat event, statistically significant mean temperature differences of -5.8 K for surface temperature and -1.1 K for air temperature were simulated. Cooling effects are more pronounced during daytime for surface temperature especially on unshaded areas, while cooling effects for air temperature are strongest during nighttime. Model results also show that the entire air volume in the study area is cooled down due to this adaptation measure, even in areas of the domain where no surfaces have been unsealed in the scenario design. The more realistic GGP scenario shows cooling effects of a comparable magnitude as the extreme GGP scenario. Thus, even partial GGP unsealing is an effective adaptation measure for reducing extreme temperatures in cities if water availability is not limited.

## 1. Introduction

Urban areas are particularly affected by climate change effects such as heat, droughts or flash floods from high precipitation intensity. The frequency, duration and intensity of extreme events has significantly increased during the past decades, and negative consequences for urban dwellers will significantly increase in future (EYRING et al., 2021; KLEEREKOPER et al., 2012). The overheating of urban areas can be attributed to radiation and heat trapping by the urban structures, the high energy storage capacity of building and surface materials, the low albedo of many built surfaces, and a reduced evapotranspiration, infiltration and water storage capacity due to surface sealing (TSOKA et al., 2020; PARKER, 2010). The high percentage of dark and impervious surfaces, typically between 24% to 45% of city areas, leads to high radiation absorption and low evaporative cooling (NWAKAIRE et al., 2020). The demand for new infrastructure and housing results in an increasing trend of sealed surfaces which is expected to continue in the next decades in many European agglomerations (WILKE, 2022). Therefore, potentials of climate change adaptation measures in cities have increasingly been investigated in recent years (BALANY et al.,



2020; TSOKA et al., 2020). As the temperature of sealed surfaces is up to 20 °C higher than that of the surrounding areas (NWAKAIRE et al., 2020), unsealing strategies are a central adaptation approach. To reconcile the requirements for climate change adaptation and urban development, strategies are needed to increase infiltration and evapotranspiration as well as decrease shortwave radiation absorption and at the same time do not require additional space (MULLANEY & LUCKE, 2014). Compared to asphalt or concrete roads and pavements, unsealed areas can increase the surface albedo and allow for evaporation

or evapotranspiration if vegetated (KOUSIS & PISELLO, 2023). Thus, the partitioning of the radiation balance into sensible and latent heat flux will be shifted towards latent heat flux, reducing the sensible and ground heat flux (DEL SERRONE et al., 2022). Furthermore, unsealed urban surfaces increase infiltration and enable a higher storage of water in the city for longer time periods to reduce flooding and drought effects. This buffering effect for extreme heat and flood improves the urban microclimate through more and longer-lasting evapotranspiration from unsealed soil water storages. Adaptation potentials and

thermal effects of unsealing measures depend on many factors such as their size, structure or physical surface properties (SEIFEDDINE et al., 2023). The area available to implement adaptation measures such as unsealing surfaces is largely limited by urban structural constraints, development and traffic usage, especially in densely populated cities (MULLANEY & LUCKE, 2014). Thus, adaptation potentials of unsealing approaches must be assessed based on the given local conditions to achieve the best possible cooling effects during heat and droughts. Grass grid pavers (GGPs) are an unsealing measure, which on the

one hand enables evapotranspirative cooling, and on the other hand can still be used for traffic, walking, parking or other activities. GGPs are a form of evaporative, porous and vegetative pavements, which increase water storage capacity and latent heat flux, and also have a higher reflectivity and emissivity than conventional urban surfaces (NWAKAIRE et al., 2020; QIN, 2015; PELUSO et al., 2022). GGPs not only show cooling effects for surface temperature but also for the surrounding area (HUANG & CHEN, 2020; SANTAMOURIS, 2013). In addition to their temperature-regulating function, GGPs can also increase

water availability for the surrounding vegetation like street trees (FINI et al., 2017; MULLANEY & LUCKE, 2014), and reduce storm water flow through a decrease in surface flow due to a higher surface roughness, infiltration and depression storage. Thus, GGPs contribute to reducing peak discharges and prevent flooding. In addition, they help improving water quality by filtering pollutants (BEAN et al., 2007). GGPs are even suitable for sub-optimal locations with slopes of over 10 % (PANNICKE-PROCHNOW et al., 2021). There are studies that have measured the air temperature above or the surface temperature of mixed

grass and concrete surfaces and found a cooling effect in direct comparison to sealed surfaces (TAKEBAYASHI & MORIYAMA, 2009). However, GGPs have not been parameterized yet for microclimate modelling such as with the established ENVI-met model. Until now, microclimate modelling studies only represented GGPs or similar surfaces as a separate mixture of pure grass and pure concrete in a stripe or chess board arrangement. Nonetheless, cooling effects for the urban microclimate in terms of air and surface temperature, mean radiant temperature and physiological equivalent temperature (PET) were found.

Those studies using ENVI-met focused on research areas in Italy (BATTISTA et al., 2022; BATTISTI et al., 2018; PELUSO et al., 2022), Malaysia (SAITO et al., 2015; TEOH et al., 2022), China (JIA & WANG, 2021), Austria (REZK, 2021) and Switzerland (HOFFMANN & GEISSLER, 2022). There is no ENVI-met study simulating GGP effects for Germany yet. TEOH et al. (2022) used the surface layer for grass already parameterized in ENVI-met for the GGPs. REZK (2021) adjusted the albedo and root depth of the pre-parameterized grass. In the study conducted by BATTISTI et al. (2018), alternating grass and concrete strips

were implemented in the model domain to roughly approximate the characteristics of GGPs. Simulated cooling effects of such implementations were more pronounced at the surface than in the atmosphere, and narrow side streets with a width between 6 and 9 meter show the strongest cooling effects (SAITO et al., 2015). JIA & WANG (2021) also found that the location of adaptation measures can have an important influence on the cooling potential. Unshaded places such as large squares showed clearer cooling effects than shaded areas like street canyons (BATTISTI et al., 2018). According to BATTISTA et al. (2022), the

installation of GGPs on a large square in Rome showed even higher cooling effects than a simulation with a high-albedo surface material or the implementation of shading measures. In many of the previously mentioned studies, GGPs were not analyzed as a single adaptation measure in the model setup but are combined with other strategies. Thus, isolated direct cause



and effect relationships between GGPs and temperature cannot be determined (TEOH et al., 2022). Our literature review shows that there is a high research interest in cooling potentials of GGP unsealings, but until now, no combined parameterization of GGPs has been developed for microclimate modelling. Modelling of GGPs in ENVI-met has only been carried out sporadically by a conceptual implementation separate grass and pavement arrangements, but GGPs have never been parameterized in any study. Thus, microclimate modelling of parameterized GGPs in ENVI-met to analyze cooling effects and adaptation potentials in dense urban environments represents a novelty. To fill this research gap, this study presents a parametrization of GGPs based upon in-situ measurements. The suitability of the parameterization is tested by analyzing the effects of unsealing surfaces by GGPs on the urban microclimate by scenario analyses using the high-resolution physically-based 3D ENVI-met model. Therefore, are parameterized model domain for a high-density residential research area in the city of Cologne/Germany was used to simulate a 20-year heat event in summer 2022. The study area is particularly exposed to heat stress. In the past, an urban heat island (UHI) effect of up to 10 K was observed in Cologne (LANUV, 2013). Climate change projections show, that not only the frequency and intensity of heat events is expected to increase in future, but also the magnitude and recurrence probability for flooding in the city (EINGRÜBER & KORRES, 2022). The model simulations are driven by meteorological measurements from our research-grade station in the study area. The model is calibrated and the model results are validated using a setup densely-distributed and quality-controlled microclimate sensor network within the study area (EINGRÜBER et al., 2022a). To evaluate the effects on air temperature and on surface temperature, GGPs are parameterized based on field measurement campaigns in the study area and implemented in the setup model domain according to the given spatial constraints. In relation to a simulation of the current sealed status-quo, two scenarios are assessed: 1) GGP implementation for all sealed areas in the model domain to identify the maximum cooling potential, and 2) usage compatible GGP implementation only in private spaces and low-traffic areas (courtyards, little frequented side streets, parking areas) while the lanes of all main roads remain sealed to identify the realistic cooling potential. Model simulation results of the current situation are compared to the two unsealing scenarios with respect to changes in the simulated air and surface temperature using statistical analyses and significance tests to reject the hypothesis that the implementation of GGPs has no significant microscale cooling effect.

## 2. Methods

### 2.1 Study area and ENVI-met model setup

Model simulations are performed for a 16-ha study area located in the southern part of the inner city of Cologne/Germany (EINGRÜBER et al., 2021). The study area can be classified as development types 2 and 5 (compact to open medium-high buildings) according to the local climate zone (LCZ) classification. Overall, around 20 % of the study area is characterized by green infrastructure such as the Volksgarten park or gardens in inner courtyards of the building blocks. Two main traffic axes run through the study area. The Vorgebirgsstraße in northeast-southwest direction and the Volksgartenstraße in a northwest-southeast direction. The lanes of Volksgartenstraße are spatially divided by a double-avenue of trees with an unsealed footpath. Furthermore, there are smaller side streets, parking areas as well as front gardens of various sizes in the study area (EINGRÜBER et al., 2024c). A 3D-gridded model domain with 1m spatial resolution was developed and parameterized for the study area using the ENVI-met model (Version V5.1.2) based on field observations and remote sensing data (EINGRÜBER et al., 2022b). In this way, real urban environment is represented in the model domain of this physically-based, spatially distributed and continuous time series model with a temporal resolution of 1 second and a spatial resolution of 1 m. The model is driven by measurements from a meteorological station (Campbell) in the urban park to define the forced lateral boundary conditions. The model performance was evaluated using sensitivity analyses (EINGRÜBER et al., 2022b). A model validation using field measurements from a densely-distributed network of 39 quality-controlled NETATMO sensors within the study area showed a high accuracy with a mean Nash Sutcliffe Model Efficiency Coefficient (NSE) of 0.94 for different weather conditions (EINGRÜBER et al., 2023b). To analyze the cooling potentials of GGPs, a 72-hour simulation of an extreme heat event was run



using this validated model. The period from 18th to 20th July 2022 represent the three hottest consecutive days of the year with
a maximum temperature of up to 40.14 ° C. This event represents a 20-year heat event for Cologne based on historical
measurements and a fitting to a Gumbel extreme value distribution (EINGRÜBER et al., 2023a). This heat event was
characterized by very low wind speeds of maximum 0.386 m/s and represents the beginning of a longer extreme drought period
in the region.

## 2.2 Parameterization of grass grid pavers

In the ENVI-met Database Manager (DBManager), a new soil profile consisting of different soil and surface materials was
parameterized to represent GGPs. Typical GGPs consist of 8 cm thick concrete stones (HOFFMANN & GEISSLER, 2022; HUNT
& COLLINS, 2008; ICPI, 2020; LIN et al., 2013; STARKE et al., 2011). A soil profile of GGPs according to Figure 1 is
implemented in ENVI-met. This requires a mixed parameterization of the concrete and substrate as well as a separate
parameterization of the grass growing in the gaps, as ENVI-met does not enable a direct mixing of vegetation and surface
materials. Furthermore, sand and gravel are used as bedding layers for the soil profile. As in the entire study area, sandy loam
is the predominant natural soil type, which is already parameterized in the database. In the same way, parameters for sand are
available in the database, but these had to be adjusted according to the GGP construction materials catalog by HOFFMANN &
GEISSLER (2022) by adjusting the thermal conductivity and heat capacity density. For the typical gravel soil below the GGPs,
the heat capacity and the thermal conductivity were parameterized according to HOFFMANN & GEISSLER (2022). The hydraulic
conductivity for gravel ranges between $10^{-1}$ and $10^{-3}$ m/s, depending on the literature source, which is why a value of $10^{-2}$ is
assumed (DAS, 2010; FREEZE & CHERRY, 1979; SHACKELFORD, 2013). For the water content of a gravel soil, the values of the
coarsest-grained parameterized sand soil are assumed. For the parameterization of the GGPs itself, the heat capacity and
conductivity were taken from HOFFMANN & GEISSLER (2022). The saturated hydraulic conductivity, the water content at
saturation and the albedo were measured in-situ within the study area for the database parameterization. The matrix potential,
the water content at field capacity and wilting point as well as the mixing coefficient water and turbidity are derived according
to the substrate in the concrete grid paver and then included as a percentage for the proportion of soil. The roughness length
for concrete is 0.010 m according to the DBManager and 0.015 m for all natural soils except sand. Due to the edge between
concrete and the substrate, the $Z_0$ for the GGP is slightly higher than that of pure concrete. Thus, the roughness was also
measured in the field. Grass is already parameterized in ENVI-met and was adapted to the characteristics of grass growing in
GGPs. Therefore, the grass height and the Leaf Area Density (LAD) were determined in the field and adjusted according to
the ratio of concrete to grass.

### 2.2.1 Field measurements of the GGP parameters

In-situ measurements of saturated hydraulic conductivity, soil moisture at saturation and albedo were conducted in the northern
part of the study area in a GGP parking lot area at Vondelstrasse 37 in spring 2023. In addition, the GGP dimensions and the
height and LAD of the grass were measured. The substrate in the gaps was sampled to be analysed in the laboratory. The
substrate-concrete ratio of the GGPs was determined in the field (39 % substrate to 61 % concrete) is used to compute the
combined GGP parameterization of the mentioned parameters taken from the literature or measured on site (see Figure 2). The
saturated hydraulic conductivity of the GGPs was determined using infiltration measurements in the parking lot. The
infiltration rate was measured using three double-ring infiltrometers. The double-ring infiltrometers consist of two concentric
stainless steel infiltration rings which are filled to equal level with water to avoid lateral flow. The vertical infiltration flux is
measured in the inner ring (EIJKELKAMP, 2012). As it is not possible to drill the rings into the soil on concrete such as GGPs,
the rings were sealed with clay on the ground to prevent lateral leakage. Measurements were carried out at five different test
sites on different parking lots in the study area. Water levels for the five tests were measured every 10 minutes. The final and
constant infiltration rate was recorded as saturated hydraulic conductivity after the infiltration rate became constant for at least



three consecutive measurements. With 1.2, 1.1, 1.1, 0.8 and 1.3 cm per 10 minutes, an average constant infiltration rate of 1.1 cm within 10 minutes was found for the GGPs, which corresponds to 66 mm/h. This results in a saturated hydraulic conductivity of 18.3 m/s*$10^{-6}$ to be used for the parameterization in the ENVI-met DBManager. Soil moisture at saturation of the substrate within the GGPs was measured at four test sites on the GGP parking lots with six repetitions each (24 measurements in total) using a calibrated ThetaProbe ML2x probe connected to an HH2 Moisture Meter and given as

volumetric soil moisture content (vol.%) calculated from the changes in the dielectric constant of the soil with an oscillation frequency of 100 MHz. Prior to the measurements, the substrate was fully saturated with water. With 32.4, 27.8, 29.7 and 28.9 % on average for each of the four test sites, an overall average soil moisture at saturation of 29.7% was found for the GGP substrate. As the substrate only covers 39% of the area of the GGPs, and the pavers itself are not permeable (saturation soil moisture 0%), the value for the combined GGP parameterization was calculated as a weighted mean. Thus, a mixed soil

moisture at saturation of 0.116 m³/m³ was used in the ENVI-met DBManager. The shortwave albedo of the GGPs was determined using a pyranometer (Vernier, 2012). The radiation was measured alternately for the incoming solar radiation and the outgoing reflection from the GGPs in a height of 1.5 meters above ground level. The albedo is calculated as the division between the incoming and outgoing radiation. The measurements in the study area were conducted during a dry period (several days without any precipitation) at a day with clear-sky conditions in spring 2023. Ten measurement repetitions have been

implemented. The albedo varied between 0.133 and 0.177 with an average of 0.144 to be used for the GGP parameterization in the ENVI-met DBManager. These measurements represent a more realistic albedo of GGPs exposed to environmental influences after some years of installation in contrast to the albedo values (0.20 - 0.25) published in the literature which are based on new (lighter) GGPs after factory production (Battisti et al., 2018; Hoffmann & Geissler, 2022; Peluso et al., 2022). In order to address the soil parameters as precisely as possible, substrate samples were taken from paver gaps and dried

at 105 °C in the laboratory to fully dry the soil material to be sieved. A 0.063 mm sieve was used to separate the sand from the silt and clay. A mixture of 71 % sand and 29 % silt-clay was found which corresponds to loamy sand according to the soil type classification. Thus, loamy sand is used as soil material for the parameterization in the DBManager. To define the $Z_0$ roughness length for the GGP parameterization, a profile measurement was conducted. As description of the extent to which the surface deviates from a completely flat surface by elevations, the number of edges between substrate and concrete and their height

were measured in the field similar to the procedure in Santos & Julio (2013). 14 substrate/concrete edges are given per profile meter with a height difference of 0.5 to 4.0 cm each (1.5 cm on average) resulting in a $Z_0$ of 0.21 m for the GGP parameterization in the DBManager of ENVI-met (see Figure 2).

### 2.2.2 Implementation of the GGP parameterization in the soil profile

The parameters for the parameterization of the three soil materials making up the GGP surface (GGPs, sand and gravel) were

directly measured in the field, calculated from the substate-concrete ratio or taken from literature are given in Table 1. These individually developed soil and surface material parameterizations were then combined to a soil profile according to the structure given in Figure 1. The final soil profile thus consists of an 8 cm thick GGP layer, which is the combined parameterization of substrate and concrete. This material is defined as a natural material to enable water flow and transportation within the material in the model. This layer is followed by a 2 cm layer of the new parameterized sand and a 10 cm bedding

layer of the new parameterized gravel, followed by the ENVI-met DB parameterization for sandy loam as natural standing substrate in the study area (see Figure 3). This soil profile was defined as non-irrigated with an assumed emissivity of 0.9. For grass growing above the GGP layer, the grass profile of the database was modified according to field measurements of the grass growing in the GGPs of the parking lots in the study area. The grass height was determined on site in representative GGPs by measuring the grass stems growing therein and calculating the mean value. The mean height of the grass in the

database was adjusted accordingly to 0.045 m, and the mean root depth was changed to 0.053 m. For the Leaf Area Density (LAD) of the grass, the average length of the grass stems with their side stems was multiplied with the average width of the

stems to calculate the grass density per GGP gap. As there are 49 grass gaps per m², a resulting LAD of 0.9702 m²/m³ was observed for ideally overgrown GGPs. The LAD is assumed to be uniform for all z-height levels of the grass (see Figure 4).

**2.3 Scenario design**

In this study, two scenarios with GGP implementation are compared to a reference run (Figure 5). Simulation 1 (S1), the reference run, describes the actual status of the study area where nearly all surfaces in the urban development are sealed by impermeable asphalt or concrete pavement surfaces, and unsealed grass areas can only be found in the urban park, road medians, front gardens of houses or back gardens in inner courtyards of building blocks. Simulation 2 (S2) represents an extreme scenario in which all sealed surfaces were replaced with GGPs. This scenario aims at quantifying the maximum effect

to be expected due to GGPs for the given meteorological conditions. Simulation 3 (S3) is intended to represent a more realistic scenario in the transition between complete sealing and a complete GGP implementation. For this scenario, the limitations of the ground surface and the actual situation in the study area were taken into account. In this usage compatible scenario, GGP unsealings were only implemented on private space and low-traffic areas like inner courtyards, little frequented side streets or parking areas while the lanes of all main roads remain sealed as in the reference run. This more realistic scenario encompasses

18,333 GGP grid cells compared to the extreme scenario with 19,958 GGP grid cells. With 0.9702 m² of grass per m³, this results in around 77,453 m³ of additional green space in S2 and around 71,147 m³ in S3. This means that there is 8.2 % less green space in S3 compared to S2. GGPs are not designed to carry high traffic loads and the higher roughness causes inconvenience for light vehicles and pedestrians (MORETTI et al., 2019; PELUSO et al., 2022). It would therefore be advisable to install them in side streets, on pedestrian pathways, in inner courtyards or in parking lots (MANTEGHI & TASNEEM, 2020).

Therefore, for the design of the realistic scenario S3 with a usage compatible GGP implementation, the lanes of high-traffic roads were kept as asphalt surfaces. While GGPs were set for all side streets, the lane width of the main traffic axes Volksgartenstraße (double-avenue in the middle of the street) and Vorgebirgsstraße was measured to determine the number of sealed grid cells in the model domain. For an assumed minimum sealed lane width of 4 meters for the main traffic roads, corresponding polygons which are not changed to GGPs are created in QGIS and implemented in the model domain INX file

using ENVI-met Monde. For all other sealed areas, the predominant asphalt and concrete sealings were replaced by GGPs for the scenario design (see Figure 5).

**2.4  Statistical evaluation methods**

Descriptive statistical analyses and significance tests are performed to test these hypotheses:

I) The applied GGPs in the scenarios S2 and S3 do not show a significant microscale cooling effect on surface temperature

and air temperature in comparison to the reference run S1.

II) The cooling effect of GGPs is not significantly higher at the surface than in the atmosphere.

III) There are no significant differences of the cooling effect for the scenarios between day and night.

IV) The cooling effect of GGPs is not significantly lower on unshaded areas in relation to shaded areas.

For the analyses, the surface temperature and air temperatures at 1, 3 and 5 m height above ground level of all GGP pixels of

all scenarios were extracted from the EDT/EDX model output files converted into a NetCDF format. To separate nighttime and daytime hours for the third hypothesis we defined 8 a.m. to 6 p.m. as daytime because the first GGP pixels are sunny at 8 a.m. and the last ones at 6 p.m. based on the shadow flag parameter. Nighttime is defined accordingly. For the fourth hypothesis, the shadow flag parameter was used to extract shaded and unshaded areas. The output data of the corresponding pixels were extracted using Python in Leonardo DataStudio. For comparability reasons between the three simulations, exactly

the same GGP pixels of S2 were also evaluated for S1 and S3. In this way, data of 19,958 pixels were extracted for all 72 simulations hours for each simulation with Python to be compared. Hourly mean values of all pixels were calculated using R



as well as frequency distributions and descriptive statistical parameters, namely mean, median, variance, minimum and maximum. Box plots were generated for intercomparison of the three simulations. Differences in mean temperature values were checked for statistical significance using a t-test. A significance level of 0.05 was assumed for all statistical tests. The same test procedure was also performed for the selected nighttime and daytime data as well as for the shaded and unshaded daytime pixels to compare the effects of shaded GGPs with GGPs directly exposed to irradiation. Additionally, NetCDF model output data was loaded into QGIS in order to map mean differences between the scenarios. The hourly layers per day were averaged using the raster calculator for individual days and the entire simulation period. By subtracting the average maps of S2 and S3 from S1, maps were produced which show the spatial variability of the mean change of temperature. To analyse, if GGPs not only show a cooling effect on air temperature but also lead to an increase in thermal outdoor comfort despite relative humidity and reflected, secondary radiation might increase due to the installation, thermal comfort indices are calculated. To quantify thermal outdoor comfort for human organisms, the biometeorological indices UTCI (Universal Thermal Climate Index) and PET (Physiological Equivalent Temperature) are determined based on the model outputs of the three simulations for all atmospheric grid cells in the model domain using BIO-met software. To quantify the effect of GGPs on perceived temperature and heat stress reduction, the absolute difference in UTCI and PET between the reference run S1 and the scenarios S2 and S3 is mapped for the entire study area on 72-hour average in the same way as mentioned above for air temperature.

### 3. Resulting cooling effects of the parameterized GGP scenarios

### 3.1 Spatial variability of temperature differences

Cooling effects for surface temperature of -2.00 K up to -8.26 K can be identified for all grid cells where GGPs have been implemented. Figure 6 maps the absolute difference in surface temperature between the reference run S1 and the extreme scenario S2 as a mean value per pixel for the 72-hour period. In built-up areas as well as in areas which are already unsealed in the reference run like the Volksgarten park or other vegetated areas, hardly any temperature differences can be detected. On average for all surface grid cells of the entire study area, a cooling effect of -3.00 K in surface temperature was found. For individual hours such as 1 p.m. of the hottest day (19[th] July 2022), the surface temperature was decreased by up to -20.01 K for single GGP grid cells. Figure 7 shows the absolute difference in 1-meter air temperature between the reference run S1 and the extreme scenario S2 as a mean value per pixel for the 72-hour period. In general, temperature differences are much less pronounced than on the ground surface with changes of -0.19 up to -2.73 K. The areas with GGPs also show clearer cooling effects, but cooling effects can be found throughout the entire study area and thus also in areas where no GGPs have been implemented such as the urban park or courtyard gardens. This means that the air volume of the entire study area is cooled down in this GGP scenario. The areas in the northern part of the Vorgebirgsstraße have a noticeably higher temperature difference. On average for all 1-m height atmosphere grid cells of the study area, a cooling effect of -0.92 K for air temperature was found. For individual hours like for 12 a.m. of the hottest day (19[th] July 2022), the surface temperature was decreased by up to -7.01 K for single grid cells in the model domain. In Figure 8, differences between surface temperature and air temperature are compared for the three individual days of the heat event. All days show cooling effects for surface temperature and air temperature. On the hottest day, 19[th] July, the strongest cooling effects can be observed with up to -9.04 K for the surface temperature and up to -5.13 K for 1-meter air temperature. On 18[th] July, weaker cooling effects can be found under street trees and in the avenue, while on 19[th] July, stronger cooling effects occur around the trees. Similarly, weaker effects are given on 20[th] July. Although the strongest cooling effects are given for the hottest day 19[th] July, the cooling effects are a little weaker for 20[th] than for 18[th] despite the 20[th] being hotter than 18[th]. This can be attributed to a limitation of the cooling effects of the GGPs due to a lack of plant-available water (EINGRÜBER et al., 2023c). As the difference of the soil water content to the field capacity (FC) continuously increases over the three days, the available water for transpiration of the GGPs decreases over time and causes smaller cooling effects on the third day with a deficit in relation to FC of up to -90% in relation to the first day



where no soil water content deficit in relation to FC was given. unshaded, tree-free areas on the third day. Although there was a decrease over time, the water content of all GGPs was still sufficient for plant evaporation throughout all three days. The soil
water content of GGPs decreased much less in shaded areas like under street trees.

### 3.2 Surface temperature differences

The boxplots in Figure 9 illustrate the distribution of surface temperatures for the three different scenarios based on hourly values calculated from the average of all GGP pixels. For S1, a large interquartile range (IQR) of 13.96 °C can be observed. With 28.54 °C, the median is higher than in the GGP scenarios (23.1 °C for S2). S2 and S3 also show a smaller IQR of 9.27
°C and 9.47 °C. For S2, a maximum temperature difference of -11.12 K can be observed on 19[th] July at noon. On average, the surface temperatures are 5.76 K cooler in S2 and 5.27 K cooler in S3 than in S1 based on the averages of all GGP pixels (72 values). These average cooling effects of GGPs on surface temperature are statistically significant according to the applied t-test for both scenarios S2 and S3 in relation to S1, but more significant for the extreme scenario S2. In Figure 10, an hourly timeseries of the mean surface temperature is given for the three scenarios. Maximum surface temperature differences between
the scenarios can be observed during the hottest hours of the day between 12 a.m. and 7 p.m., while the smallest temperature differences were simulated between 7 p.m. and midnight. The deviation of the dataset into day and night is illustrated in the boxplots of Figure 11. During night hours, smaller deviations in both the maximum and average temperature values for both scenarios S2 and S3 can be found. On average, daytime cooling effects are 4.4 K higher than at night for S2 (4.09 K more for S3). While significant differences were found between the different scenarios both during day and night, p-values are lower
during daytime and thus, cooling effects are stronger and more significant for daytime. The deviation of the dataset into shaded and unshaded areas is represented in the boxplots of Figure 12. The differences in surface temperature are also significantly higher on unshaded areas than on shaded areas. In scenario S2, a temperature difference of up to -13.43 K can be observed on unshaded areas, whereas this is only -10.85 K in shaded areas. The higher temperature differences on the unshaded areas can primarily be observed in the reduction of the maximum temperatures of the days. While significant surface temperature
differences were found between the different scenarios both for shaded and for unshaded areas, p-values are lower for unshaded areas and thus, cooling effects are stronger more significant for direct sunlit surfaces.

### 3.3 Air temperature differences and thermal outdoor comfort

While cooling effects of GGPs on surface temperature are highest during daytime, cooling effects on air temperature are stronger during nighttime. The boxplots in Figure 13 illustrate the distribution of air temperatures for the three different model
runs based on hourly values of all atmosphere grid cells for 1 m, 3 m and 5 m height above ground level. Air temperature reacts less sensitive to the installation of the GGPs. All boxplots show similar scattering with a similar IQR and standard deviation. The reference run S1 has the warmest temperatures at every height level, and the extreme GGP scenario S2 shows the lowest temperatures while the realistic scenario S3 ranges in between. The difference in the cooling effects between the scenarios is strongest at a height of 1 m and decreases with increasing distance to the ground surface. Thus, cooling effects are
larger in lower height levels. The strongest cooling effect of -2.89 K on air temperature occurs at 1 m height level on 19[th] July at 11 a.m. for scenario S2. On average, air temperature is 1.08 K cooler based on the averages of all GGP pixels (72 values). In Figure 14, an hourly timeseries of the mean air temperature is given for the three scenarios and the three height levels above ground surface. During the coldest and warmest hours of the days, the temperatures are highest at 5 m altitude in S2 and S3 and lowest at 1 m altitude, while for S1, temperatures are sometimes slightly higher at 1 m altitude and smallest in 5 m altitude,
and sometimes hardly any differences of the temperature with height can been observed for S1. Thus, temperature increases with height for S2 and S3 while decreases for S1. Greater air temperature differences between the scenarios can be observed mainly during the coldest hours of the day between 1 a.m. and 7 a.m. The deviation of the dataset into daytime and nighttime is illustrated in the boxplots of Figure 15 exemplarily for a height level of 1 m above ground surface. It becomes clear, that the



differences between the scenarios are more pronounced at nighttime. According to the median, there is a slightly higher cooling

effect during the nights. In summary, the analysis revealed significant temperature differences at the surface for S2 and S3, with maximum values of -11.12 K/-10.33 K, which are most pronounced during the day on unshaded areas. The temperature differences in the air showed cooling effects of up to 2.88 K/2.67 K, which are highest closer to the surface and during nighttime. The sensible heat flux and the soil heat flux are reduced due to the GGP implementation, while the sensible heat flux, relative humidity and soil water content are decreased due to the unsealing, the increased LAD and the different material

properties of the GGPs. To analyse, if the air temperature cooling effect of GGPs also leads to an increase in thermal outdoor comfort despite relative humidity and reflected, secondary radiation increases due to the GGPs, thermal comfort indices have been calculated. A map of the absolute difference in the Universal Thermal Climate Index (UTCI) between the reference run S1 and the extreme scenario S2 is shown in Figure 16. It becomes clear, that thermal comfort is significantly increased due to the GGP implementation in the entire study area, especially in the street canyons by up to -2.6 K. On spatial average, an UTCI

improvement of -1.7 K was found. In S2, the perceived temperatures on the roads with GGPs drop from very strong heat stress to only strong heat stress, and also the Physiological Equivalent Temperature (PET) shows a decrease of the perceived temperature on spatial average by -4.6 K and up to -6.8 K.

Overall, we conclude that the four initially defined hypotheses can all be rejected according to the performed statistical significance t-tests, and the corresponding alternative hypotheses can be accepted as follows:

I) The applied GGPs in the scenarios S2 and S3 show a significant microscale cooling effect on surface temperature and air temperature in comparison to the reference run S1.

II) The cooling effect of GGPs are significantly higher at the surface than in the atmosphere.

III) There are significant differences of the cooling effect for the scenarios between day and night.

IV) The cooling effect of GGPs is significantly lower on unshaded areas in relation to shaded areas.

Furthermore, we found that the cooling effects are more pronounced on the hottest day of the three-day simulation period (19[th] July with fully-autochthonous weather conditions), but for all days, a temperature reduction was observed. This clearly states that even on July 20[th] where partial cloudiness has pronounced after noon, a significant heat mitigation potential can be concluded for the parameterized GGPs.

## 4. Discussion

The identified significant temperature differences across all GGP pixels indicate the overall cooling effects on average, but do not account for spatial variations within the study area. The variety of surfaces and shapes in urban areas exhibits a high diversity of features influencing the urban microclimate. The interaction and interplay of these diverse surfaces contribute to the complexity of the urban environment and must be considered. The intrinsic characteristics of surfaces are additionally influenced by external factors such as radiation, geometry, position within the flow field, and immediate surroundings (OKE,

1982). At the surface level, particularly between the Volksgarten park and the adjacent tree-lined double avenue, strongest cooling effects might occur, as combinations of various adaptation strategies usually yield the best results. In this context, both the closeby park and the tree-lined avenue, in combination with the GGPs are likely to be most effective. In a study by PELUSO et al. (2022), the combination of cool surfaces, hedges and trees achieved the best cooling effects as the air temperature could be reduced by over 3 K. BATTISTI et al. (2018) also concluded that combining GGPs with trees and green roofs leads to the

best results. In the same way, the high cooling effects along the eastern section of the avenue Volksgartenstraße and the northern part of the Vorgebirgsstraße in our study area can be explained. Numerous trees have been planted at these locations, which noticeably reduce temperatures even at heights of 5 m above ground surface. The identified significant cooling effects of GGPs on surface temperature are strongest during the hottest hours, which can be explained by the thermal properties of the GGPs. The temperature of the atmosphere, on the other hand, is indirectly reduced by the lower sensible heat flux of the



ground surface which leads to smaller cooling effects of air temperature. Measurement and simulation studies are in agreement with the magnitude of the cooling effect of the newly-developed GGP parameterization for ENVI-met in this paper, like the analyses for Rome and Fondi/Italy of BATTISTA et al. (2022) and PELUSO et al. (2022). The results from Ipoh and Malacca/Malaysia show very similar maximum surface temperatures effects, but only about half of the cooling effect for the maximum air temperature (SAITO et al., 2015; TEOH et al., 2022). It should be noted that the analyses in Ipoh only examined

the joint effect of GGPs with roadside trees. In many of the mentioned studies, it is therefore difficult to determine the isolated effect of GGPs as different adaptation measures were often combined with each other. Results for air temperature in Vienna are significantly lower than in this study, although GGPs were represented by pure grass there (REZK, 2021). A simulation in Hong Kong also concludes weaker cooling effects for the air temperature (JIA & WANG, 2021). Results cannot directly be compared due to the hotter and drier climatic conditions. At the same time, it needs to be taken into account that all of these

studies have not applied a specific parameterization for GGPs and thus, transferability is highly limited. Our findings of significant differences in the cooling effect with increasing height above ground surface is also in agreement with theoretical studies and can be explained by the mixing of warmer and cooler vortices in the air with higher altitude above ground (OKE et al., 2017). The observed significant differences in the cooling effects between day and night, which are greater during daytime in our simulations can be traced back to the fact that the sensible heat flux was reduced by approx. 130 to just around 10 W/m²

in S3. TAKEBAYASHI & MORIYAMA (2009) performed tests with various parking lot surfaces, also including concrete-grass mixtures, and compared them with each other and with an asphalt parking lot. The surface temperatures above the grass areas also showed a stronger cooling effect during daytime compared to the asphalted parking areas due to a reduction of sensible heat flux by about 100 to 150 W/m², which agrees well with the results of our study. In addition, a lower absorption of heat by GGPs can be explained by the thermal material properties. Increasing the thermal inertia and minimizing the ratio between net

radiation and heat conduction into the ground can reduce the temperatures of surfaces (WANG et al., 2021). A smaller proportion of heat is dissipated into the ground despite the higher thermal conductivity of the GGPs, as a higher proportion of energy is directly transferred into latent heat flux. In addition, the high thermal conductivity in deeper layers of the ground reduces the surface temperatures during the day. The thermal inertia of the soil can be reduced by the lower volumetric heat capacity of the GGPs resulting in a lower heat storage during the day, especially during the hottest hours (PELUSO et al., 2022).

According to GUI et al. (2007), heat conduction and heat capacity only cause a reduction in maximum temperatures. The results of our study therefore contradict the statement by MANTEGHI & TASNEEM (2020) that porous surfaces only have a marginal cooling effect when filled with soil. With the substrate assumed in our study, this also causes an average of 35 % of the cooling effect. As the strongest cooling effects occur during the midday hours, GGPs are an effective measure to minimize peak temperatures, which are particularly harmful to human health. The calculated biometeorological indices PET and UTCI prove

that thermal stress can significantly be reduced by the parameterized GGPs especially during the hottest hours of the simulated days. Although the albedo of the GGPs and the grass (0.144 and 0.2) is smaller than of the mean pavements in S1 (0.35) resulting in a higher energy input, cooling through evapotranspiration compensates and even exceeds this effect which is also proven by LEE et al. (2016) who found a cooling effect for air temperature of up to 3.4 K and an average of 1.1 K during the course of the day similar to our simulations. At nighttime, cooling effects of GGPs on surface temperature are much lower,

but significantly higher differences in air temperature occur in both S2 and S3 than during daytime. The process of heat dissipation during the night can be reduced by installing GGPs. On the one hand, the lower volumetric heat capacity results in lower storage of solar energy during the day. On the other hand, nocturnal cooling is also controlled by the process of evaporation of the surface. This means that the surface and especially the air layers close to the ground are cooled at night. This can explain why the air temperature in S2 and S3 was on average around 11 to 16 % lower during the coldest hours,

compared with the difference during the hottest hours. In a study in Basel at 4 a.m., also a lower air temperature was found over GGPs compared to other surfaces such as asphalt, concrete, stone slabs and gravel (HOFFMANN & GEISSLER, 2022). In the study by TAKEBAYASHI & MORIYAMA (2009), more significant temperature differences were also found for air temperature



at nighttime. GUI et al. (2007) also observed a higher reduction in minimum air temperatures than in maximum air temperatures. In agreement to our determined significant differences in the cooling effects between shaded and unshaded GGP

areas being more pronounced over sunny areas, higher temperature differences on unshaded surfaces have also been found in BATTISTI et al. (2018). A study on Mancini Square in Rome also demonstrated that the GGPs have the strongest cooling effects especially in the center of the square without any shadings. Locations at the edge of the square next to a building and in an adjacent street under trees showed significantly lower cooling effects (BATTISTA et al., 2022). The positioning of the GGP is therefore of central importance. On unshaded surfaces, lower p-values, and thus greater temperature differences, were found

in our study as more energy is absorbed at unshaded areas which leads to higher temperatures. A greater reduction in temperature can then be achieved by installing GGPs. On the other hand, grass growing on unshaded areas can die earlier during extreme drought events. This could cause smaller cooling effects on unshaded areas in relation to shaded areas when the evapotranspiration of the dead grass reaches zero, and due to the smaller albedo of that surface in contrast to the sealed surfaces of S1, all irradiated energy is then transferred into sensible and ground heat flux. Soil water content has already

decreased slightly within the three simulated days and caused smaller cooling effects on the third day. After several days without any precipitation and artificial irrigation, the effect of evaporative cooling could therefore become negligible on unshaded areas, while grass can survive for longer periods on shaded surfaces like under street trees with higher water availability. Our results also clearly showed that water availability is higher in the direct vicinity of trees. Irrigation of GGPs during particularly dry phases could therefore maintain the cooling function permanently. The combined implementation of

GGPs and urban trees can make GGPs more resilient even during prolonged drought periods. Some studies also showed that permeable pavements in combination with urban trees improve water availability for both trees and GGPs (FINI et al., 2017; MULLANEY & LUCKE, 2014). Overall, it was shown in this study that the cooling effects of the GGPs are highest on the hottest simulation day 19$^{th}$ July and higher for the extreme scenario S2 than for the more usage-compatible scenario S3. Also, GGPs not only cool down the air above the surfaces where GGPs have been implemented, but air temperature was also decreased in

other parts of the study area where no GGPs have been set in the scenarios like in the urban park or in inner courtyard gardens. Thus, the GGPs are able to cool down the air volume of the entire study area. Besides these significant cooling effects and other benefits such as water filtration and storage, it should not be neglected that GGPs also present particular challenges. These include the potential for damage if traffic volumes are too high, as the stones are not designed to bear high loads, and the reduction in infiltration performance due to compaction and clogging of the pores after some time (PANNICKE-PROCHOW

et al., 2021). Therefore, GGPs require regular maintenance. GGP albedo can also be further reduced while aging. Another key challenge is the death of the grass in extremely dry and hot regions, which is why the implementation of GGPs should always be examined on a site-specific basis (MULLANEY & LUCKE, 2014). Furthermore, accessibility for cyclists, pedestrians and people with walking disabilities must be guaranteed (TAKEBAYASHI & MORIYAMA, 2009). In reality, further sealed strips would be needed for users such as cyclists, and possibly a partial sealing of side streets would still be necessary. Thus, the more

usage-compatible scenario S3 is still not a realistic one. But although S3 includes 8.2 % less GGP implementation than S2, cooling effects are only slightly smaller. This demonstrates, that there is no linear relationship, and even smaller GGP implementations (percentages) can have significant cooling effects in comparison to S1. Particular attention should be paid to water availability in the overall context of neighborhood planning. In terms of water management, GGPs can be a central strategy for flood protection measures due to increased infiltration and water storage for reducing surface runoff and thus peak

flows (BEAN et al., 2007). The results of this study prove the suitability of the newly-developed parameterization of GGPs for microclimate modelling in ENVI-met to fill the identified research gap. The simulations showed that partial unsealing with GGPs is a suitable climate change adaptation measure. Full unsealing scenarios would have even stronger cooling effects. Even higher adaptation potentials can also be expected when combining GGP unsealings with further technical solutions and nature-based solutions like with blue or green roofs (EINGRÜBER et al., 2023d; EINGRÜBER et al., 2024). Overall, our findings

can have important implications for decision-making in urban planning aiming to mitigate future heat stress, droughts, flooding



and improve thermal outdoor comfort. Thus, these climate adaptation pathways can contribute to several sustainable development goals (SDGs) of the United Nations (UN), especially the goals 3, 11 and 13.

**5. Conclusion**

As GGPs have never been parameterized for microclimate modelling with ENVI-met before, a new parameterization was developed using in-situ measurements to fill this research gap. Based on measurements of saturated hydraulic conductivity with double-ring infiltrometers, of soil moisture at saturation point using FDR probes, of surface albedo using pyranometers, as well as many other measurements of the substrate and vegetation of GGPs, a new database profile for ENVI-met was parameterized. To analyse the cooling potential of the GGPs, scenario analyses were performed for an urban high-density study area in Cologne/Germany using a validated ENVI-met model. An extreme scenario with hypothetical GGP installation 465 on all sealed surfaces, and a more usage compatible scenario where GGPs were not installed on main traffic roads were implemented in the model domain to investigate the microclimatic effects of this parameterization. The GGP unsealings are highly effective in mitigating urban heat stress in the entire city quarter to adapt to the negative effects of anthropogenic climate change as they significantly reduce the surface and air temperature. During the hottest hours, differences of up to -20.1 K were found for surface temperature, while differences of up to -7.1 K were identified for air temperature in 1 m height above ground 470 level. On spatial average for the entire model domain, cooling effects of up to -11.1 K for surface temperature and up to -2.9 K for 1 m air temperature were simulated during this 20-year heat event in Cologne in summer 2022. On temporal average for the 3-day heat event, statistically significant mean differences of -5.8 K for surface temperature and -1.1 K for 1 m air temperature were concluded. Cooling effects are more pronounced during daytime for surface temperature especially on unshaded areas, while cooling effects on air temperature are strongest during nighttime as the GGPs store less thermal energy 475 during the day and therefore emit less into the atmosphere at night. While surface temperature is only decreased in areas with GGP installations, air temperature indicates that the entire air volume in the study area is cooled, even in areas of the model domain where no surfaces have been unsealed in relation to the reference run like in the urban park or inner courtyard gardens. The cooling effect of the GGPs on air temperature decreases with the distance from the ground surface as a cooling source. As the more usage compatible GGP scenario shows cooling effects of nearly the same magnitude as the extreme GGP scenario, 480 even partial GGP implementations represent an effective adaptation measure for temperature regulation in dense urban environments. Based on the model outputs, it was also found that the thermal material properties of the GGPs cause about one third of the surface temperature differences, while the evapotranspiration is the main cooling process driver. Within the study area, a high spatial variability of cooling effects was found. Thus, adaptation potentials of GGPs must be assessed by urban planners based on the given local conditions to achieve the best possible cooling effects including factors such as radiation, 485 geometry, position within the flow field, and immediate surroundings. Our results also showed that GGPs not only reduce temperatures, but also increase thermal outdoor comfort as the indices PET and UTCI also quantify a significant reduction of heat stress for humans in the study area. The adaptation potential of the GGPs is largely limited by water availability. Our simulations demonstrated that the effect of evaporative cooling reduced during the 3-day heat period, especially in unshaded areas, while water availability is higher in the direct vicinity of street trees. Thus, in further research, combinations of this new 490 GGP parameterization with other technical and nature-based adaptation strategies should be investigated. Especially a combination of GGPs with street trees could be a reliable approach to increase water availability for both during extreme, prolonged heat and drought periods to increase resilience by reducing urban heat stress and health risks in a changing climate.

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





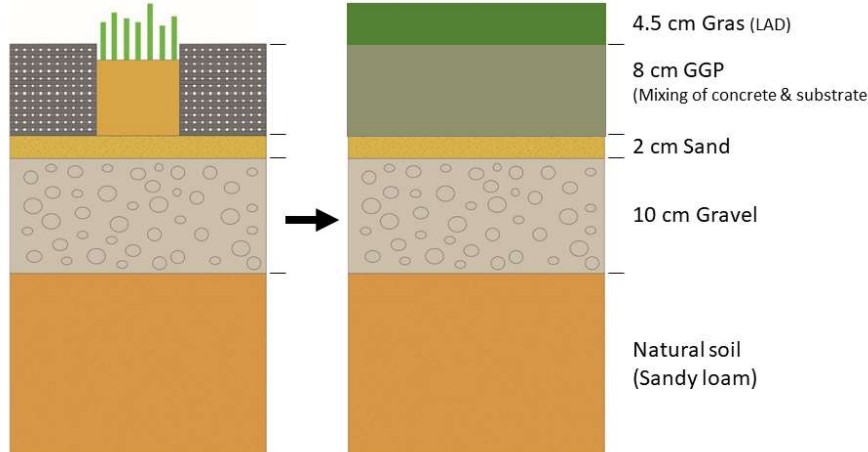

**Figure 1: Schematical structure of a GGP on the left, and developed implementation in ENVI-met on the right (own illustration based on HOFFMANN & GEISSLER 2022).**

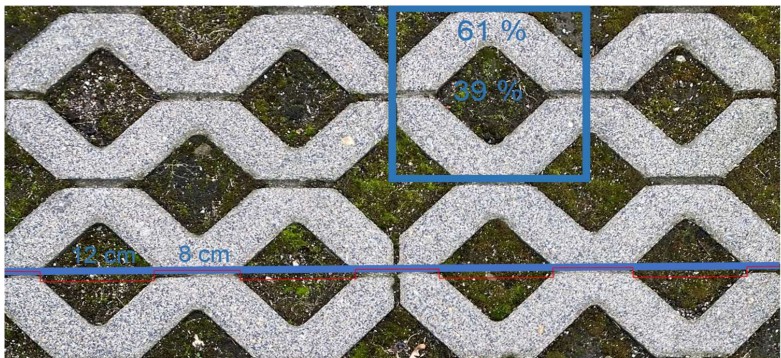


**Figure 2: Dimensions of the grass grid pavers measured in the study area and location of the profile for roughness determination.**

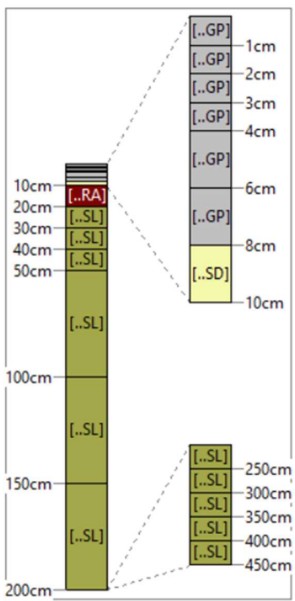

**Figure 3: Vertical structure of the GGP soil profile in the Database Manager; SL = Sandy Loam, RA = Gravel, SD = Sand, GP = Grass Grid Pavers (own illustration in ENVI-met DBManager).**



**Table 1: Parameterization of the soil materials for the construction of grass grid pavers in the Database Manager; FC = Field Capacity, WP = Wilting Point, LS = Loamy Sand.**

| Parameter | Grass Grid Paver | Sand | Gravel |
|---|---|---|---|
| Type of Material (Definition) | Natural material (Water flow is occurring) | Natural soil (Water flow is occurring) | Natural soil (Water flow is occurring) |
| Water content at saturation/FC/WP [m³(Water)/m³(Soil)] | 0.11578 / 0.0585 / 0.02925 (Own measurement, FC+WP: DB Manager LS*0.39) | 0.395 / 0.135 / 0.0068 (DBManager Sand) | 0.395 / 0.135 / 0.0068 (DBManager Sand) |
| Matrix potential [m] | -0.0351 (DB Manager LS*0.39) | -0.121 (DBManager Sand) | -0.121 (DBManager Sand) |
| Hydraulic conductivity [m/s*10⁻⁶] | 18.3 (Own measurement) | 176 (DBManager Sand) | 10000 (DAS, 2010; FREEZE & CHERRY 1979; SHACKELFORD, 2013) |
| Volumetric heat capacity [J/(m³K)*10⁻⁶] | 1.54 (HOFFMANN & GEISSLER 2022) | 1.463 (DBManager Sand) | 1.28 (HOFFMANN & GEISSLER 2022) |
| Clapp Hornberger Constant b [dimensionless] | 1.7082 (DB Manager LS*0.39) | 4.05 (DBManager Sand) | 4.05 (DBManager Sand) |
| Thermal conductivity [W/mK)] | 2.0 (HOFFMANN & GEISSLER 2022) | 1.6 (HOFFMANN & GEISSLER 2022) | 0.7 (HOFFMANN & GEISSLER 2022) |
| $Z_0$ Roughness length [m] | 0.21 (Own profile measurement) | | |
| Albedo [fraction] | 0.144 (Own measurement) | | |
| Emissivity [fraction] | 0.9 (DBManager Sandy Loam/Concrete) | | |
| Mixing coefficient Water [m²/s] | 0.001 (DBManager Sandy Loam/Concrete) | | |
| Turbidity Water [1/m] | 2.1 (DBManager Sandy Loam/Concrete) | | |

Database-ID: [1XXCSG]
Name: Grass 4,5 cm aver. dense (Cologne Südstadt)
Color: ▬

| Parameter | Value |
|---|---|
| Alternative Name | (None) |
| CO2 Fixation Type | C3 |
| Leaf Type | Gras |
| Albedo | 0.20000 |
| Emissivity | 0.97000 |
| Transmittance | 0.30000 |
| Plant height | 0.04500 |
| Root Zone Depth | 0.05300 |
| Leaf Area (LAD) Profile | 0.97020,0.97020,0.97020,0.97020,0.97020,0.97020,0.97020,0.97020,0.97020,0.97020 |
| Root Area (RAD) Profile | 0.10000,0.10000,0.10000,0.10000,0.10000,0.10000,0.10000,0.10000,0.10000,0.00000 |
| Season Profile | 0.80000,0.80000,0.80000,1.00000,1.00000,1.00000,1.00000,1.00000,1.00000,0.80000,0.80000,0.80000 |

**Figure 4: Parameterization of the grass growing above the GGPs (own illustration in ENVI-met DBManager).**



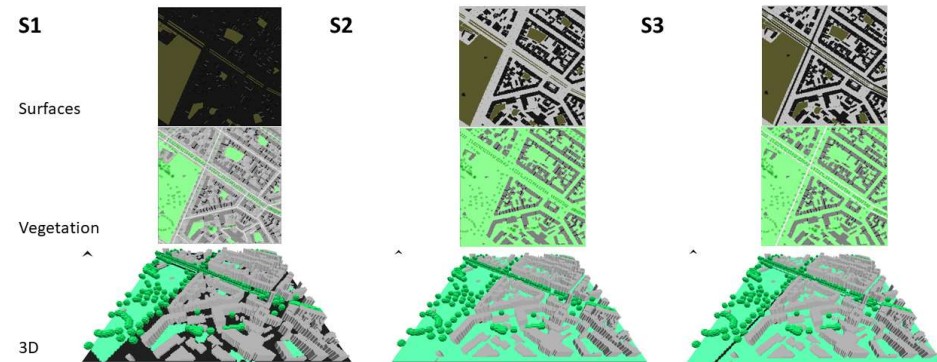


**Figure 5: Representation of the surfaces, vegetation and 3D model domain for the designed scenarios (S1 = reference run without any GGPs, S2 = extreme scenario: GGP implementation on all sealed surfaces, S3 = realistic scenario: usage compatible GGP implementation on low-traffic areas while lanes of the main traffic roads Volksgartenstraße (W to E) and Vorgebirgsstraße (N to S) are still sealed).**

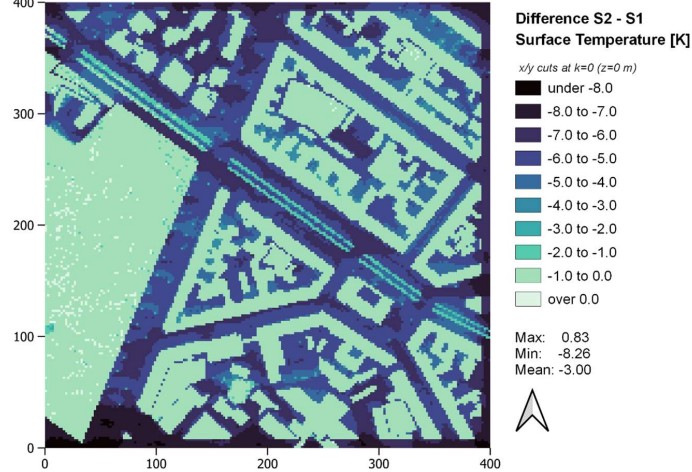


**Figure 6: Spatial distribution of the mean differences in surface temperature [K] between S1 (reference run) and S2 (extreme scenario) over the entire 72-hour simulation period (18th - 20th July 2022). The x and y axes indicate the spatial distance in the study area.**

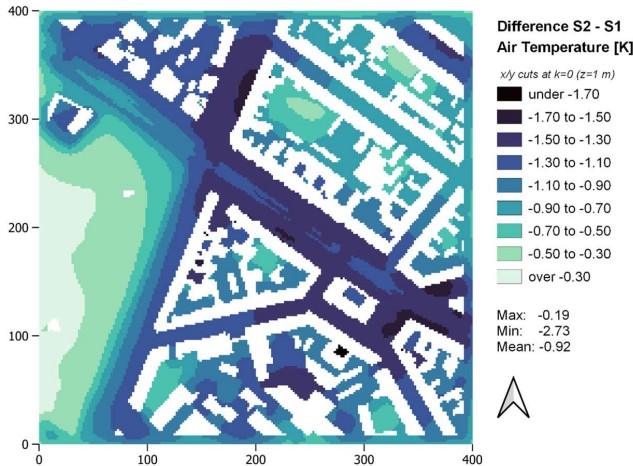



**Figure 7: Spatial distribution of the mean differences in 1 meter height air temperature [K] between S1 (reference run) and S2 (extreme scenario) over the entire 72-hour simulation period (18th - 20th July 2022). The x and y axes indicate the spatial distance in the study area.**

**18.07.2022 Surface Temperature [K]**  Max: 0.00  Min: -7.85

**18.07.2022 Air Temperature [K]**  Max: -0.20  Min: -2.10

**19.07.2022 Surface Temperature [K]**  Max: 2.24  Min: -9.04

**19.07.2022 Air Temperature [K]**  Max: -0.23  Min: -5.13

**20.07.2022 Surface Temperature [K]**  Max: 1.39  Min: -8.88

**20.07.2022 Air Temperature [K]**  Max: -0.08  Min: -2.04

**Figure 8: Spatial distribution of the mean differences in surface temperature [K] (left) and air temperature [K] (right) between S1 (reference run) and S2 (extreme scenario) for all three single days of the study period. The color scales correspond to those in Figure 6 and Figure 7. The x- and y-axes indicate the spatial distance in the study area.**



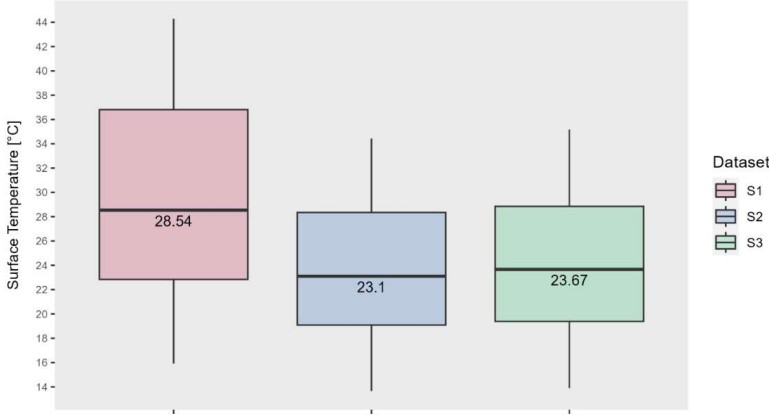

**Figure 9: Boxplots of the mean surface temperature for the three simulations S1, S2 and S3. Averages of all GGP grid cells for the 72 hourly output values.**

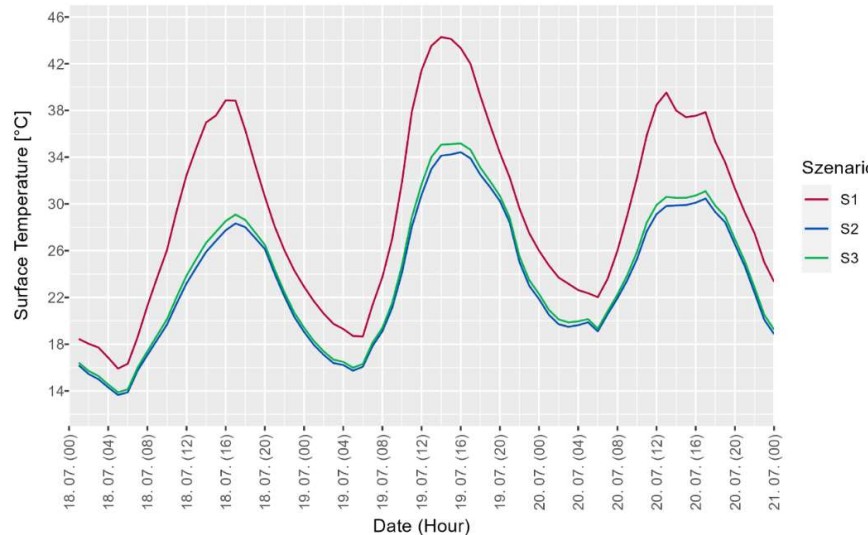


**Figure 10: Time series of the mean surface temperature of all GGP grid cells in the model domain for the simulations S1, S2 and S3.**

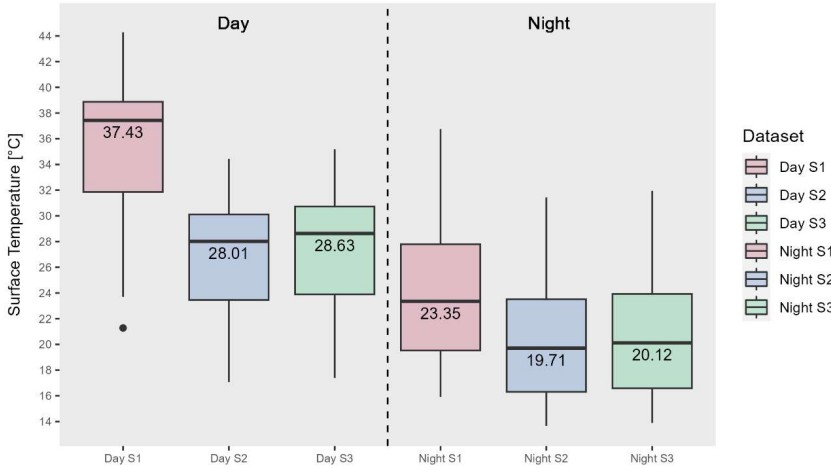



**Figure 11: Boxplots of the mean surface temperature for the three simulations S1, S2 and S3 divided into daytime (left) and nighttime (right). Averages of all GGP grid cells for the 72 hourly output values.**

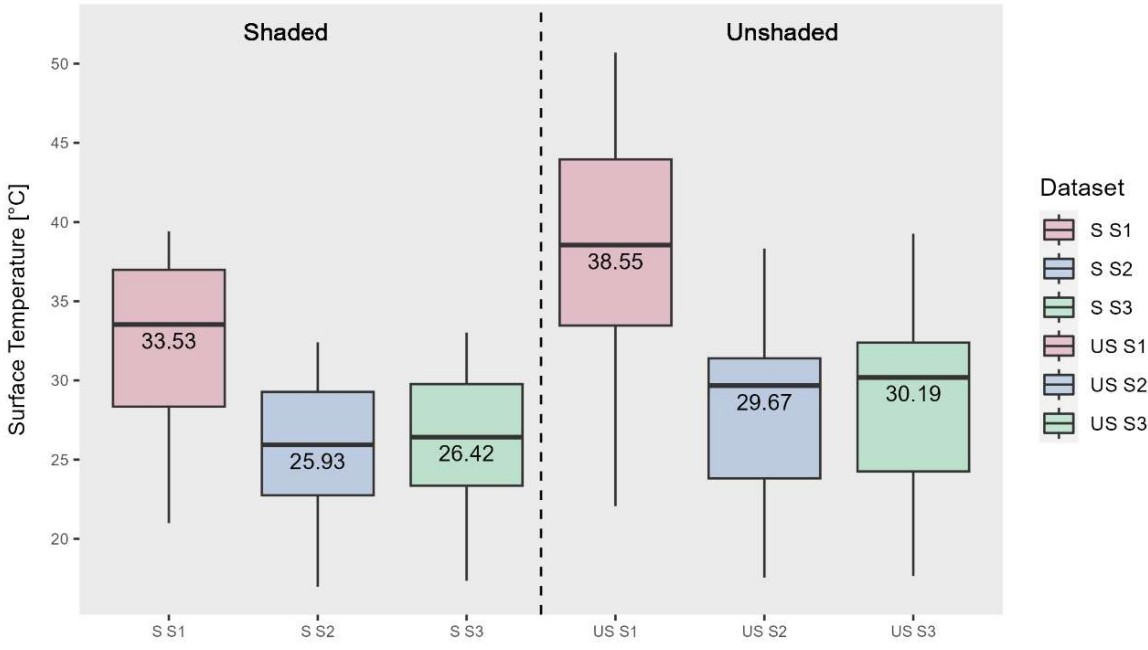


**Figure 12: Boxplots of the mean surface temperature for the three simulations S1, S2 and S3 divided into shaded areas (left) and unshaded areas (right). Averages of all GGP grid cells for the 72 hourly output values.**

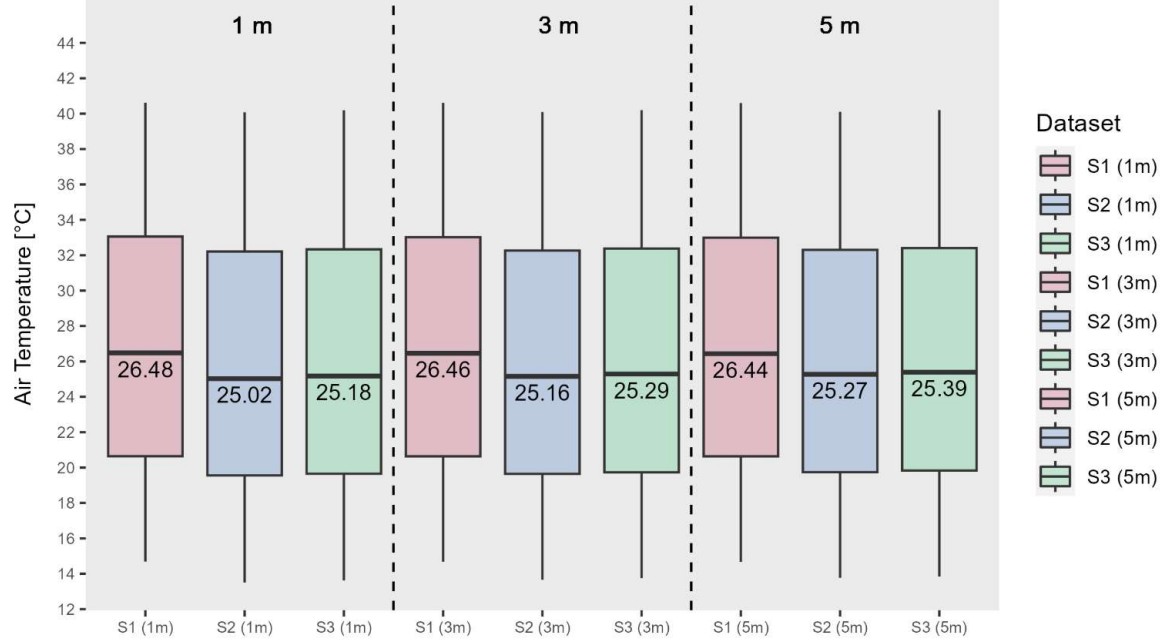

**Figure 13: Boxplots of the mean air temperature for the three simulations S1, S2 and S3 in 1 m height above ground level (left), 3 m height above ground level (middle), and 5 m height above ground level (right). Averages of all GGP grid cells for the 72 hourly output values.).**







**Figure 14: Time series of the mean air temperature of all atmosphere grid cells) in the model domain in 1 m height above ground level (solid lines), in 3 m height above ground level (dashed lines), and in 5 m height above ground level (dotted lines) each for the simulations S1, S2 and S3.**




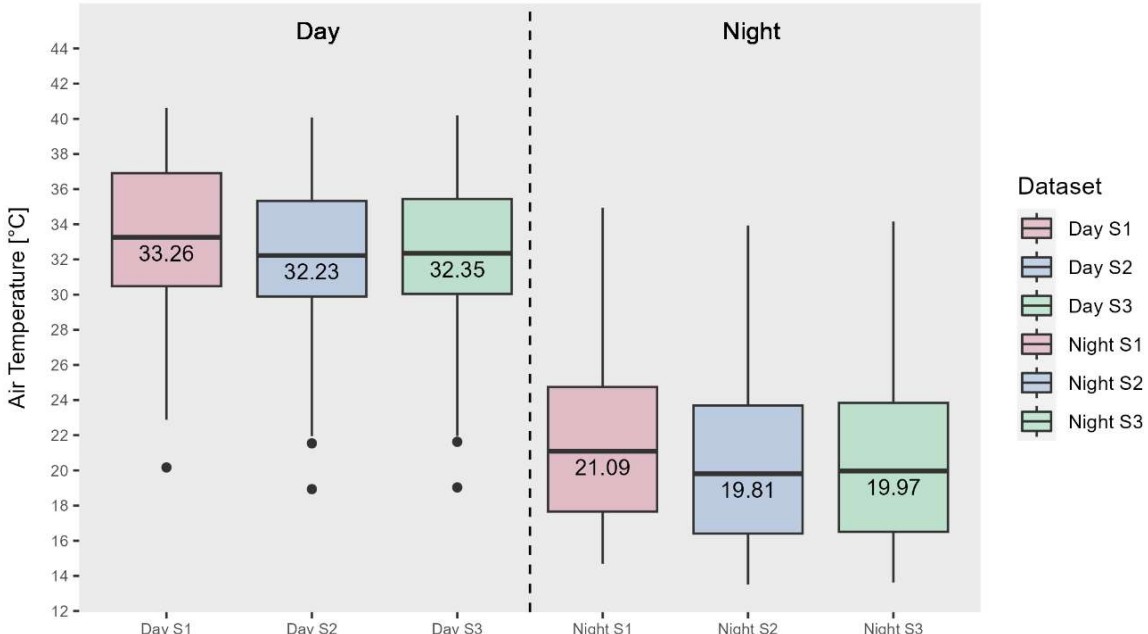

**Figure 15: Boxplots of the mean air temperature in 1 m height above ground level for the three simulations S1, S2 and S3 divided into daytime (left) and nighttime (right). Averages of all GGP grid cells for the 72 hourly output values.**

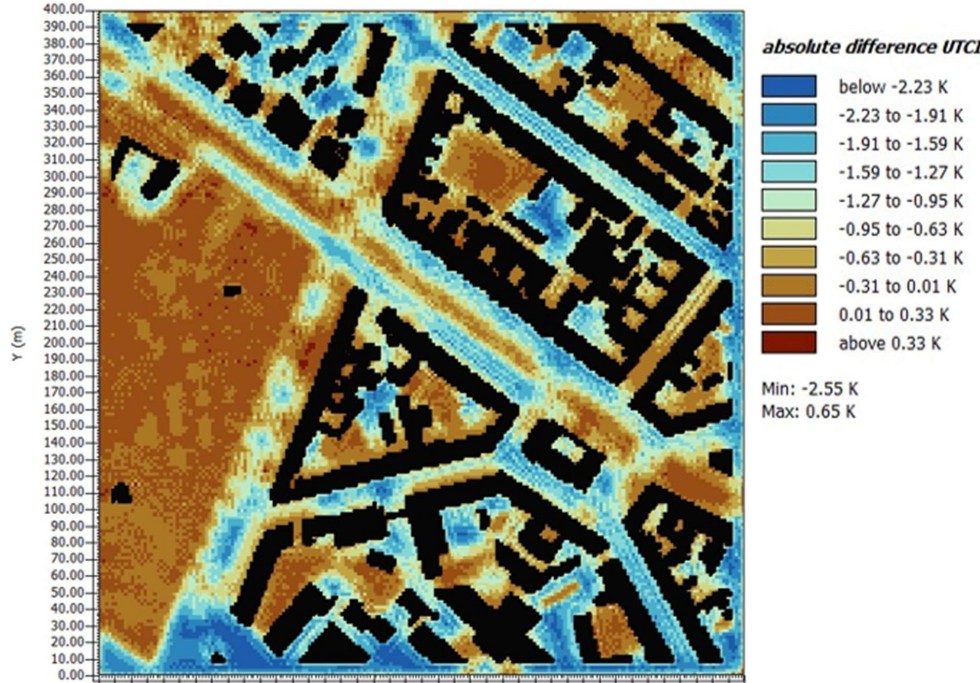

**Figure 16: Absolute difference in the biometeorological Universal Thermal Climate Index (UTCI) between the reference run S1 and the extreme scenario S2 in a height of 3 m above ground level.**



**Code/Data availability**

The exact version (V 5.1.2 Compatibility Release) of the ENVI-met model used for this paper, the parameterizations and the input data to run the ENVI-met model and to reproduce the model outputs of all the performed scenario simulations shown in this paper, as well as all the scripts for post-processing of the model output data needed to derive the presented results of this manuscript are permanently archived as supplementary material files on the Zenodo repository (EINGRÜBER et al., 2024b).

**Author contributions**

NE developed the model domain and setup the ENVI-met microclimate model for the study area. AD performed the field measurements in the study area for the GGP parameterization, and statistically analysed the simulation outputs. KS and WK supported in the implementation of the modelling concept and data analyses. NE and AD prepared the manuscript draft. KS and WK participated in writing and supervising the work. All have read and accepted the manuscript for submission.

**Competing interests**

The contact author has declared that none of the four authors has any competing interests.

**Disclaimer**

Publisher's note: Copernicus Publications remains neutral with regard to jurisdictional claims in published maps and institutional affiliations.

**Acknowledgements**

We would like to thank all citizen scientists and local cooperation partners within the study area who make it possible to carry out our measurements in private spaces. We acknowledge partial financial support for the project from the Gesellschaft für Erdkunde zu Köln (GfE), and financing of the Article Processing Charge from the DFG (German Research Foundation, grant no. 491454339).

**Financial support**

We acknowledge partial financial support for the project from the Gesellschaft für Erdkunde zu Köln (GfE), and financing of the Article Processing Charge from the DFG (German Research Foundation, grant no. 491454339). This open-access publication was funded by Universität zu Köln.

**500-character short summary**

Climate change adaptation measures like unsealings can reduce urban heat stress. As grass grid pavers have never been parameterized for microclimate model simulations with ENVI-met, a new parameterization was developed based on field measurements. To analyse the cooling potential, scenario analyses were performed for a densely-developed area in Cologne. Statistically significant average cooling effects of up to -11.1 K were found for surface temperature, and up to -2.9 K for 1m air temperature.