# Peer review of "Simulation of the heat mitigation potential of unsealing measures in cities by parameterizing grass grid pavers for urban microclimate modelling with ENVI-met (V5)"

_EGUsphere, 2024_

## Referee Comment (RC2)

**Review on egusphere-2024-697:**

"Simulation of the heat mitigation potential of unsealing measures in cities by parameterizing grass grid pavers for urban microclimate modelling with ENVI-met (V5)"

Eingrüber *et al.*

Submitted to GMD

*2024-08-14*

I think that the proposed parameterization of GPPs is a useful tool, as it allows to better map the influence of surface characteristics on urban heat. However, the paper could be better written and structured. I have added a lot of general and technical comments as I think there are some major limitations that come with this manuscript, such as citation errors, imprecise writing and redundancy, drawing conclusions from missing analysis results, and more. Therefore, I would rate the manuscript with a major revision.

**General Comments**

- The overall impression of the work is particularly affected by imprecise writing und missing structures. There are many redundancies and the manuscript would be greatly improved by adding paragraphs and clearer/more precise writing. This needs to be done throughout the manuscript, not just the technical comments.

- Please ensure that all programs and concepts (such as UTCI) have a reference and all measurement advices are described properly with model, company, …

- Use abbreviations for air temperature ($T_a$) and surface temperature ($T_s$).

- I could only find t-test results for $T_s$, not $T_a$! Why? I think this is a major shortcoming, as you mention in the Abstract that you found a statistically significant difference for $T_a$, but never checked for it? Anyway, did you check for autocorrelation? Isn't it an assumption of the t-test that the data points of the samples must be independent? A three-day diurnal cycle must have some autocorrelation.

- Surface and air temperature are important measures for evaluating GPPs. However, I don't understand why the focus is on surface temperatures rather than thermal indices, as I think "heat mitigation potential", as stated in the title, is better assessed by thermal indices. In

addition, Figure 16 shows the difference in UTCI at 3 m between S1 and S2, but is UTCI at 1 m not better for assessing human outdoor heat stress? And what is about the difference between S1 and S3? Later, I will ask if it is possible to combine some figures (see Point 2 in **Tables and Figures)**. I wonder if such a figure would be possible for UTCI?

- When evaluating heat mitigation potential, how exactly did you analyze UTCI and PET changes (compare also to my comment on Line 340)? When and how often did the UTCI classes changed from very strong heat stress to strong heat stress? Pixel-wise or on spatial average? For individual hours or on temporal average? I think it is important to provide detailed information on this.

- Only pixels where sealed surfaces have been replaced by GPPs are compared (Lines 244-245). This is fine to investigate the effect of GPPs alone, but shouldn't there also be a comparison of all pixels to see the overall effect of GPPs on the outdoor thermal comfort of a whole neighborhood at pedestrian level? The surroundings are otherwise neglected, right? I would suggest to analyses $T_a$, $T_s$, and all other indices with the same approach. Therefore, I think it is better to compare all pixel (exclude buildings).

- A lot of self-citation of conference abstracts, which do not seem to hold necessary information. Although I think it is ok to cite conference abstracts when no other information is available, citing this many conference abstracts without relation is not appropriate.

**Specific Comments**

**1. Introduction**

- Re-structure the introduction with paragraphs (e.g., in lines 37, 54, 88 …) and give each paragraph a subject such as general characteristics of urban areas and the impact of climate change, the need for adaptation measures, introduction of GPP and the research done about it, and the contribution of your paper to this field. These would improve the readability.

- As this study focuses on heat adaptation, please focus more on the effect of GGPs on heat. It is relevant to mention the effect of GGPs on urban hydrology, as you do in lines 59-64. However, I think this is sufficient to do it shorter to focus on urban heat. E.g. remove lines 93-95.

**2. Methods**

- It could be useful to have a figure with the study area and land cover classes, locations of the weather station, and Netatmo network, indicating the main roads and backyards, etc… and using this figure to describe the characteristics of the study rather than citing conference abstracts which does not always hold the information.

- Same as for the introduction: Please re-structure in different paragraphs such as "Model domain", "Forcing and evaluation data (measurements)" and "Model setup".
- After reading this chapter, I don't think that I could reproduce the ENVI-met setup. I also did not find much information in the cited abstracts and papers (only in Eingrüber 2023b). Or add information like: "More information on the study area and the model step up can be found in …". But ensure that this information is present in the cited paper.
- Section 2.2: Please restructure. It does not get quickly apparent what is explained when. Lines 130-151 give an overview of the entire parametrization process, but also information about values used in DBManager. It seems a bit mixed up. In section 2.2.2 you write about field measurements of LAD (Lines 202-208). However, section 2.2.1 is about field measurements. I think this could be structured more clearly, also to remove redundancy.

**3. Results**

- Please add a table with all t-test results.
- Are there somewhere results for PET? Maybe add them to the Appendix.

**4. Discussion**

- The discussion is comprehensive. However, it is not always clear whether $T_a$ or $T_s$ is being discussed. As with the other sections, please restructure and add paragraphs to make it clearer (e.g., day-night, scenarios, shaded-unshaded, model limitations, …).

**Technical Comments**

**1. Introduction**

- Line 19: Period is missing.
- Line 64: Studies or study? Only one Citation.
- Line 90: physics-based / numerical
- Line 91: "*Therefore,* **a**…" not "are"?
- Line 93-95: I thought the focus is on heat, not on precipitation and flooding?
- Line 97: "*using* **data from** *a densely-distributed*"
- Line 104: "*air and surface temperature***s"**
- Line 110: Add citation for LCZ.
- Line 115: 1 m
- Line 116: Add citation for ENVI-met model.

- Line 116: You cite a conference abstract (Eingrüber et al., 2022b) without any information on how the 3D data of your model domain was obtained.
- Line 119: Campbell? Please provide more information on the measurements, its instruments etc. Or provide a citation where everything can be found.
- Lines 120-123: What are the results of the validation (Eingrüber et al., 2022b)? A conference paper is cited without any results. I think this sentence/citation is not necessary, as in the next sentence a Journal article is cited and information about the accuracy is provided.
- Lines 120-124: Model evaluation of what? Air temperature?
- Line 122: I think in Eingrüber et al., 2023b the NSE is given with 0.91?
- Line 125: °C
- Line 126: I did not find any information about this Gumbel distributions in the conference abstract of Eingrüber et al. 2023a. Is this of relevance for the ENVI-met modelling? Or only for the 20-year heat extreme? I don't understand why this citation is added as there is no relation?
- Line 140: add units
- Line 146: roughness length = $Z_0$ in line 148?
- Line 156: … *"(39 % substrate to 61 % concrete)* **and** *is used"* ?

**2. Methods**

- Line 174: Sometimes you have a space between Number and %, sometimes not.
- Line 175: New paragraph
- Line 176: Model, Company?
- Line 179: "… **on** *a day with clear-sky…"*
- Lines 184-192: New paragraph and maybe re-arrange entire section with soil characteristics after hydraulic conductivity at line 175.
- Line 188: "*As description of the extent to which the surface deviates from a completely flat surface by elevations…"* Which surface? The GPP surface?
- Line 190/191: How is the profile meter defined? Does the orientation of the profile meter play a role? Or is it simply an approximation?
- Line 196: "parameters" instead of "parameterizations" (Compare Line 194)
- Line 201: Why emissivity of 0.9? References?
- Line 212:
- Lines 226.228: "*While GGPs were set for all side streets, the lane widths of the main traffic axes Volksgartenstraße (double-avenue in the middle of the street) and Vorgebirgsstraße* **were** *measured to determine the number of sealed grid cells in the model domain."*
- Line 244: References for Python and Leonardo DataStudio.
- Line 246: Reference for R.

➤ Lines 252-254: I don't think you have to describe how you created all results such as how you calculated the difference of mean temperature.

➤ Line 253: "*The hourly layers per day were averaged using the raster calculator for individual days and the entire simulation period.*" What does "hourly layers per day" mean?

➤ Lines 254-266: Rephrase.

➤ Lines 257-258: References for PET, UTCI, and BIO-met.

➤ Line 259-261: Not necessary.

**3. Results**

➤ Line 264: "*surface temperature**s** from -2.00 K up to -8.26 K…*"

➤ Line 268: Is this now the comparison of all grid cells or only the grid cells with GPPs? (compare with line 244).

➤ Lines 264-…: Please always define if it is air or surface temperature ($T_a$ & $T_s$), not only "temperature".

➤ Line 277: "*…(19th July 2022), the surface temperature was…*" not sure about, but do you mean $T_a$?

➤ Line 284: Move to discussion.

➤ Line 288: ""

➤ Line 295: "*S2 and S3  show a smaller IQR of 9.27 °C and 9.47 °C**, respectively**.*"

➤ Line 297: Maybe add a table with the statistical results (t-, and p-value) to the Appendix.

➤ Section 3.1: Please use paragraphs for $T_s$ and $T_a$.

➤ Line 303: "*(4.09 K more for S3)*". More?

➤ Line 304: " **Since** *p-values are lower during* **the** *day* **than at night** , *cooling effects are stronger and more significant* **during the day**."

➤ Line 315: Again, all grid cells? Or only the GPP grid cells as written in the figure caption?

➤ Line 320: 72 values or timesteps?

➤ Line 330: New paragraph or even section? In general, why summarize results? Isn't this a part for conclusion?

➤ Line 330-335: "*ensible*  *and*  *soil heat flux**es**  **decreased** due to  GGP implementation, while  sensible heat flux, relative humidity and soil water content  decreased due to  unsealing,  increased LAD and  different material properties of the GGPs.*"

➤ Line 335-337: Rephrase.

➢ Line 339: Does thermal comfort improve in the entire study area (Fig. 16?). It is a bit difficult to distinguish between the different classes in Figure 16. For me, it appears that a lot of pixels have a value between -0.31 K and 0.33 K, which does not seem to be the entire study area.

➢ Line 340: When and how often did the UTCI classes changed from very strong heat stress to strong heat stress? Pixel-wise or on spatial average? For individual hours or on temporal average?

➢ Lines 343-353: Move to Conclusion?

**4. Discussion**

➢ Lines 360-364: Surface or air temperature? Not clear, as first it is written "At the surface level", but then it is compared to $T_a$.

➢ Line 383: Again, which differences are higher during the day? I think its $T_s$, right?

➢ Line 387: "*The surface temperatures  **of** the grass areas*"

➢ Lines 384-388: For my understanding: In your study the sensible heat flux was reduced by 130 Wm$^{-2}$, which means from 140 Wm$^{-2}$ to 10 Wm$^{-2}$, and in TAKEBAYASHI & MORIYAMA (2009) it was reduced from 250 Wm$^{-2}$ by 100 Wm$^{-2}$ to 150 Wm$^{-2}$. Is this correct? But isn't there a big difference between 10 Wm$^{-2}$ and 130 Wm$^{-2}$? Is there an overview of all fluxes?

➢ Lines 299-401: If I understood correctly, you wrote about $T_s$ in the adjacent sentences. Here you mention thermal indices, which are less affected by $T_s$ but rather by $T_a$. So, I don't get the context. Or is this already part of the following sentences, where you start writing about $T_a$?

➢ Line 404-405: Rephrase: "*At nighttime, cooling effects of GGPs on surface temperature are much lower, but significantly higher differences in air temperature occur in both S2 and S3 than during daytime.*"

➢ Line 409: $T_a$ increase in percentage? I think this is rather uncommon. It does not seem that you use the Kelvin scale? To my understanding, everything else will give arbitrary results, as Celsius has no "absolute zero" point in 0°C. So, its maybe better to avoid using percentage for temperatures changes.

**Tables and Figures**

➢ Table 1: "*Thermal conductivity [W/(mk)]*"

➢ Figures 9, 11, and 12: Merge to one figure with subplots a, b, and c? Would make understanding easier. Then you would need only one legend with S1, S2, and S3. In addition, a boxplot of all $T_a$ and $T_s$ differences would be nice to see, not averaged by timestep, just 72x400x400 values to get an impression of the entire "cooling" distribution.

➢ Same for figures 13 and 15.

➢ Can this merged figure also be done for UTCI?

- ➢ Why is figure 10 so small compared to figure 14? You could also merge them. Could you add confidence intervals? As you only show the spatial mean.
- ➢ Figure 8 caption: "*The color scales correspond to those in Figure 6 and Figure 7.*" Is it possible to add these color scales to the figure 8? Makes it easier to analyze.
- ➢ Figures 6-8: I wonder about the edges of almost all plots. Also, in comparison to Figure 5. It somehow looks like the model domains have different edges between S1 and S2/S3 (Figure 5)? Does this lead to the high differences at the edges from Figure 6?
- ➢ Figure 16: What about the mean?

---

## Referee Comment (RC3)

**Review Report for the Manuscript ID egusphere-2024-697**

September 2, 2024

**Summary of the Study**

The manuscript titled "Simulation of the heat mitigation potential of unsealing measures in cities by parameterizing grass grid pavers for urban microclimate modelling with ENVI-met (V5)" presents an innovative approach to urban heat mitigation through the application of grass grid pavers (GGPs). In light of the increasing frequency of summer heat events and the growing urban population, the study highlights the need for climate adaptation measures that do not require additional space or alter the fundamental function of urban areas.

The authors have developed a new GGP parameterization for the ENVI-met V5 microclimate model, filling a research gap in the field. The study includes scenario analyses that assess the cooling potential of GGPs in a high-density urban area in Cologne, Germany, under extreme heat conditions. The realistic implementation scenario also demonstrates substantial cooling effects, confirming that partial unsealing through GGPs can effectively mitigate urban heat, provided that water availability is sufficient.

**General Opinion**

The study presented in this manuscript is timely and relevant, particularly as cities worldwide face the challenges posed by climate change and increased urban heat. The use of GGPs as a measure to reduce urban temperatures through unsealing strategies is a contribution to the field of urban climate adaptation. The development of a new parameterization for GGPs within the ENVI-met model is also an advancement. However, there are several issues that need to be addressed to enhance the manuscript's impact and suitability for publication in a journal like *Geoscientific Model Development* (GMD).

**Issues**

1. **Insufficient Development Contribution:** While the development of the GGP parameterization for ENVI-met is a key aspect of the study, it is not sufficiently highlighted. The contribution of this development to the broader field of urban microclimate modeling is underemphasized, and the manuscript could benefit from a more detailed explanation of the parameterization process, including its novelty and significance compared to existing approaches.

2. **Lack of Review on Similar Parameterizations:** The manuscript does not include a review of similar parameterizations in other microscale urban climate models, such as PALM, MITRAS, or ENVI-met itself. A thorough comparison with these models would provide context for the significance of the new GGP parameterization and could strengthen the paper's contribution to the field.

3. **Need for Discussion on Cooling Effect Measures:** The manuscript could be improved by discussing the differences in cooling effect measures, specifically air temperature ($T_{air}$), Universal Thermal Climate Index (UTCI), and surface temperature ($T_{surface}$). This discussion would benefit from referencing studies such as `https://doi.org/10.1175/BAMS-D-20-0193.1` and `https://doi.org/10.1016/j.enbuild.2023.113324`, which address the complexities of these different metrics in the context of urban heat mitigation.

4. **Reiterating Points from Reviewer Comments:** Most of the issues I recognized have already been listed by RC2 (`https://egusphere.copernicus.org/preprints/2024/egusphere-2024-697/egusphere-2024-697-RC2-supplement.pdf`). I agree with the points raised, particularly regarding the need for a more in-depth discussion of the modeling approach and its implications for the accuracy and applicability of the results.

---

## Author Response (AR1)

**Response to Reviewer Reports:**

In order to ensure quick comprehensibility and clarity for the reviewers, we have formatted the reviewer comments in italics and our answer in standard format. The resulting changes in the manuscript are given in quotation marks.

**RC #1:**

*The paper addresses an actual urban microclimate issue an adds significant information to the knowledge about colling the city by vegetation and specifically by grass grid pavers. The approach to the research is well substantiated. The methodology description is sufficiently complete to allow reproduction. The results provide a good basis for much additional research, for example the relationship with the albedo of surface materials. More suggestion for further research would be desirable.*

Thank you very much for your valuable feedback. We have now added a corresponding section with more suggestions for further research also with relation to surface albedo: "Thus, in further research, combinations of this new GGP parameterization with other technical and nature-based adaptation strategies like roof and facade greenings or sunsails should be investigated to identify potential reinforcing effects. Especially a combination of GGPs with street trees could be a reliable approach to increase water availability for both during extreme, prolonged heat and drought periods to increase resilience by reducing urban heat stress and health risks in a changing climate. It would also be interesting to analyse the microclimatic effects of GGPs which are designed using water-permeable concrete or which are made of surface materials with an even higher albedo. Further research could also analyse more specifically which of the parameters of the new GGP parameterization like the albedo of the GGP surface, the heat conductivity and heat storage capacity of the applied materials or the evapotranspiration of the GGP vegetation cause the highest percentage contribution to the overall cooling effect simulated in this study. In this way, it could also be identified if GGPs still have a cooling potential when fully dried. To optimize the performance of this climate change adaptation strategy in urban areas, further research could also intercompare the cooling effects of this GGP parameterization in our mid-latitude study area to study areas with different climatic conditions like semiarid climates, or to other cities with different urban geometries and arrangements to better evaluate the usability of GGPs for future urban planning.".

*The description is sometimes redundant, the authors should scan the paper and reduce repetitions, it would improve the quality of the presentation.*

We have scanned the paper and reduced several redundancies now throughout the manuscript. For example, we removed duplicate and therefore irrelevant method descriptions like in (old) lines 259 – 261, or redundant result presentations and summaries of findings like in (old) lines 330 – 333.

**RC #2:**

**General Comments:**

*The overall impression of the work is particularly affected by imprecise writing und missing structures. There are many redundancies and the manuscript would be greatly improved by adding paragraphs and clearer/more precise writing. This needs to be done throughout the manuscript, not just the technical comments.*

Thank you very much for your valuable feedback. We have tried to better arrange the structure of all the sections by adding paragraphs as well as by removing redundancies throughout the entire manuscript now.

*Please ensure that all programs and concepts (such as UTCI) have a reference and all measurement advices are described properly with model, company, …*

We have added all the relevant references for PET, UTCI and BIO-met now. We also added the corresponding references as well as provided the version numbers for all used software programs such as Python and R. Furthermore, all details on the technical specifications of the measurement devices including their model number and company name are described and referenced now.

*Use abbreviations for air temperature (Ta) and surface temperature (Ts).*

We have abbreviated Ta and Ts in this paper now so that it becomes clear which temperature effects are presented for all instances in the results section as well as which temperature effects are evaluated for all instances in the discussion section.

*I could only find t-test results for Ts, not Ta! Why? I think this is a major shortcoming, as you mention in the Abstract that you found a statistically significant difference for Ta, but never checked for it?*

We forgot to mention the statistically significant difference for Ta in the corresponding results section and therefore added the following sentences now: "These average cooling effects of GGPs on Ta are statistically significant at 1 m height above ground level according to the applied t-test for both scenarios S2 and S3 in relation to S1, but more significant for the extreme scenario S2. For greater heights like 5 m, cooling effects are not statistically significant (see Appendix 2)." The results of the tests are given in Appendix 2 now as also mentioned in the specific comments.

*Surface and air temperature are important measures for evaluating GPPs. However, I don't understand why the focus is on surface temperatures rather than thermal indices, as I think "heat mitigation potential", as stated in the title, is better assessed by thermal indices. In addition, Figure*

*16 shows the difference in UTCI at 3 m between S1 and S2, but is UTCI at 1 m not better for assessing human outdoor heat stress? And what is about the difference between S1 and S3? Later, I will ask if it is possible to combine some figures (see Point 2 in Tables and Figures). I wonder if such a figure would be possible for UTCI?*

As also written in the Specific and Technical Comments, we mainly want to focus on the thermal (physical) effects of the GGPs and not directly on the biometeorological comfort effects. Therefore, we only show that one figure exemplarily for UTCI as some kind of outlook. We have now also added a corresponding figure to the Appendix 1 to present the difference in PET, but do not want to add further figures for the given reason. In the title, we mention "heat mitigation" which is the physical expressions aiming to analyse the thermal effects close to the physical processes (sensible/latent heat fluxes, etc.). Thus, Ts and Ta are the most relevant resulting variables expressing the thermal effects of the processes in this physical context of heat mitigation = energy reduction. We explicitly do not mention "heat stress mitigation" of "thermal comfort" in the title nor in the aims in the manuscript text which would of course require a more comprehensive analysis of biometeorological indices for assessing thermal comfort of human organisms. So, the overall outcome of the paper is that the new GGP parameterization is physically suitable and significantly can reduce heat (and as an outlook also reduce heat stress as starting point for further research).
Regarding (old) Figure 16: Actually, the figure caption was wrong as the map already shows the difference in UTCI at 1 m height above ground level. You are absolutely right that 1 m is the most relevant height level for assessing heat load for pedestrians. We have corrected the figure caption accordingly now. All maps presented in the manuscript for Ta, UTCI and PET are equally given for 1 m height above ground level (also the new additional figure in Appendix 1).

*When evaluating heat mitigation potential, how exactly did you analyze UTCI and PET changes (compare also to my comment on Line 340)? When and how often did the UTCI classes changed from very strong heat stress to strong heat stress? Pixel-wise or on spatial average? For individual hours or on temporal average? I think it is important to provide detailed information on this.*

As also mentioned in the corresponding comment below, this expression is on temporal and spatial average. We modified the sentences accordingly to: "On temporal and spatial average for all grid cells in 1 m height above GGP pixels, an UTCI improvement of -1.7 K was found. In S2, the perceived temperatures drop from very strong heat stress to only strong heat stress on temporal and spatial average for all grid cells in 1 m height above GGP pixels. Also, the Physiological Equivalent Temperature (PET) shows a decrease of the perceived temperature on temporal and spatial average of all areas where GGPs have been implemented by -4.6 K and up to -6.8 K.". Spatial differences can be identified in the maps.

*Only pixels where sealed surfaces have been replaced by GPPs are compared (Lines 244-245). This is fine to investigate the effect of GPPs alone, but shouldn't there also be a comparison of all pixels to see the overall effect of GPPs on the outdoor thermal comfort of a whole neighborhood at pedestrian level? The surroundings are otherwise neglected, right? I would suggest to analyses Ta, Ts, and all other indices with the same approach. Therefore, I think it is better to compare all pixel (exclude buildings).*

This is correct, only GGP pixels have been compared to investigate the effect of GGPs mainly alone for two reasons: 1.) With this manuscript, we want to test the new GGP parameterization and evaluate its effects and therefore focus on GGP areas alone. 2.) We want to focus our research close to the physical processes itself (sensible/latent heat fluxes, etc.) and not analyse the spatial reach distance of the effects into other units like the urban park area, etc. This is why our statistics like the boxplots with comparisons between shaded and unshaded GGPs only include the model outputs of the GGP grid cells. Nonetheless, we additionally give the results for the entire study area of the neighborhood in pedestrian height level (all pixels 400 x 400 m excluding buildings) in this manuscript: "The areas with GGPs show clearer cooling effects, but cooling effects can be found throughout the entire study area and thus also in areas where no GGPs have been implemented such as the urban park or courtyard gardens. This means that the air volume of the entire study area is cooled down in this GGP scenario. [...] On average for all 1-m height atmosphere grid cells of the study area, a cooling effect of -0.92 K for Ta was found.", "On average for all surface grid cells of the entire 16 ha study area, a cooling effect of -3.00 K in Ts was found.". The spatial distributions of the cooling effects (and the statistics for all pixels excluding buildings) can be found in the corresponding difference maps of the entire study area (new Figure 6 for Ts, Figure 7 for Ta, Figure 12 for UTCI and Appendix 1 for PET). In further research, we want to compare these overall cooling effects of the GGPs for the entire neighborhood to the cooling potentials of other climate change adaptation measures then.

*A lot of self-citation of conference abstracts, which do not seem to hold necessary information. Although I think it is ok to cite conference abstracts when no other information is available, citing this many conference abstracts without relation is not appropriate.*

As also described in detail in several specific and technical comments below, we have now removed citations of conference abstracts from the manuscript which do not hold additional (available) information necessary for the reproduction of the methods etc. Instead of those conference abstract citations, we have now inserted corresponding citations from journal publications which hold the relevant information. We only left conference abstract citations in the manuscript now for instances where no other information is available in specific relations.

**Specific Comments:**

1. Introduction:

*Re-structure the introduction with paragraphs (e.g., in lines 37, 54, 88 ...) and give each paragraph a subject such as general characteristics of urban areas and the impact of climate change, the need for adaptation measures, introduction of GPP and the research done about it, and the contribution of your paper to this field. These would improve the readability.*

We have divided the introduction into 5 paragraphs in a similar a similar way as you suggested (one more paragraph to separate the subjects of microclimatic characteristics of GGPs and research done about it). We have decided to not write down the subjects for the paragraphs as this seems to be not the typical way for publications in GMD and sometimes interrupts causal bindings in the reading flow.

*As this study focuses on heat adaptation, please focus more on the effect of GGPs on heat. It is relevant to mention the effect of GGPs on urban hydrology, as you do in lines 59-64. However, I think this is sufficient to do it shorter to focus on urban heat. E.g., remove lines 93-95.*

You are right, we have removed the relation to precipitation and flooding of lines 93-95 now to keep it more focused on heat and instead outline this fact in relation to heat only: "Local climate change projections show, that not only the frequency of heat events is expected to increase in future, but also the intensity.".

2. Methods:

*It could be useful to have a figure with the study area and land cover classes, locations of the weather station, and Netatmo network, indicating the main roads and backyards, etc... and using this figure to describe the characteristics of the study rather than citing conference abstracts which does not always hold the information.*

The study area differentiation and the validation of the model outputs using the NETATMO sensor network was already done in previous research and is not directly relevant for the goals of this paper. As also described in the Technical Method Comments below, we have now removed the citation of a conference abstract for the corresponding instances in the method chapter, and instead better placed the citation of the relevant journal publication there (Eingrüber, N., Korres, W., and Schneider, K.: Microclimatic field measurements to support microclimatological modelling with ENVI-met for an urban study area in Cologne, Adv. Sci. Res., 19, 81-90, 2022.). In that paper, the study area and the indication of the main roads/backyards, etc. as well as the locations of the meteorological station and the NETATMO sensor network are already shown.

*Same as for the introduction: Please re-structure in different paragraphs such as "Model domain", "Forcing and evaluation data (measurements)" and "Model setup".*

The description of the model domain, model setup, forcing can be found in detail in EINGRÜBER et al. 2024d, and the validation with the measurements is provided in the paper EINGRÜBER et al. 2023b. Therefore, we have now better placed those citations to make this clearer. We have now subdivided section 2.1 into four paragraphs as you suggested so that the different methodological steps are better logically separated from each other.

*After reading this chapter, I don't think that I could reproduce the ENVI-met setup. I also did not find much information in the cited abstracts and papers (only in Eingrüber 2023b). Or add information like: "More information on the study area and the model step up can be found in ...". But ensure that this information is present in the cited paper.*

You are absolutely right that the information could not reproducibly be found in the previous version. Of course, the goal and focus of this paper is the new parameterization of the GGPs and the investigation of their cooling effects in the ENVI-met model. The ENVI-met model of the study area itself can be seen as given. Therefore, a detailed description of the model setup is not directly relevant for this paper. This paper builds up on that given model and implements the new developed GGP parameterization in that model to identify their microclimatic effects and cooling potentials. As you suggested, we have now added this information and therefore also added a paper where all the geodata and domain setup is described so that this information is correctly present: "More information on the setup of the ENVI-met model can be found in EINGRÜBER et al., 2024d.".

*Section 2.2: Please restructure. It does not get quickly apparent what is explained when. Lines 130-151 give an overview of the entire parametrization process, but also information about values used in DBManager. It seems a bit mixed up. In section 2.2.2 you write about field measurements of LAD (Lines 202-208). However, section 2.2.1 is about field measurements. I think this could be structured more clearly, also to remove redundancy.*

We have now structured section 2.2 into 4 paragraphs based on the material components of the GGP soil profile parameterization; Component 1: the three bed materials, Component: 2 GGPs itself, Component 3: the grass growing in the GGPs. In section 2.2, the general design of the GGP profile and the values which were used from literature or databases for the individual material parameterizations (thus externally enriched) are given, while in section 2.2.1, all the field measurements (internally enriched) are explained. In section 2.2.2, the externally and internally enriched data of the individually developed soil and surface material parameterizations of the three components (the three bed materials sandy loam, gravel and sand, the GGPs, and the grass) were then combined to the parameterized vertical soil profile. You are right that the representation was not clear in the previous version as some descriptions were mixed up between the sections and redundances were given. We have now restructured this to clearly separate these parts and removed redundances. The field measurements of LAD were wrongly placed in section 2.2.2 in the previous version. Therefore, we shifted this part to the field measurement section 2.2.1 now. Instead, we added a more precise explanation to 2.2.2 now: "The parameterized grass from Figure 4 is placed on that soil profile of Figure 3 to represent the entire GGP structure in the ENVI-met model domain.".

3. Results:

*Please add a table with all t-test results.*

We have now added a corresponding table to the Appendix (Appendix 2) with all t-test results of the paper for all intercomparisons for Ta and Ts like shaded-unshaded, daytime-nighttime, etc.

*Are there somewhere results for PET? Maybe add them to the Appendix.*

As also written in the Technical Figure Comments, we mainly want to focus of the thermal effects of the GGPs and not directly on the biometeorological effects. Therefore, we only show that one figure exemplarily for UTCI as some kind of outlook. We have now also added a corresponding figure to the Appendix to present the difference in PET and referenced it in the text.

4. Discussion:

*The discussion is comprehensive. However, it is not always clear whether Ta or Ts is being discussed. As with the other sections, please restructure and add paragraphs to make it clearer (e.g., day-night, scenarios, shaded-unshaded, model limitations, …).*

We abbreviated Ta and Ts for all instances in the discussion now so that it always becomes clear which temperature effect is being discussed. We also restructured and subdivided the discussion section into several paragraphs now to better separate the discussion for overall cooling effects, daytime-nighttime intercomparisons, shaded-unshaded intercomparisons, limitations and challenges of water availability, etc.

**Technical Comments:**

1. Introduction:

*Line 19: Period is missing.*

We added an information on the time of the maximum delta now: "[…] for the hottest hour of a simulated 3-day heat wave in summer 2022 […]".

*Line 64: Studies or study? Only one Citation.*

We meant "studies" but forgot to insert the second citation. We have now added the important contribution from Fɪɴɪ et al., 2017 to this sentence.

*Line 90: physics-based / numerical*

According to the model type classification of Lawrence Dingman, 2002, the term for the simulation basis is physically-based, while the solution method is of course numerical. We added the numerical solution method of ENVI-met to the instance 25 lines above.

**Simulation Basis**

***Physically Based*** Uses equations derived from basic physics [e.g., conservation of mass, energy, or momentum; force balance; diffusion (see Table 2-1)] to simulate flows and storages.

*Line 91: "Therefore, **a**…" not "are"?*

We corrected it correspondingly.

*Line 93-95: I thought the focus is on heat, not on precipitation and flooding?*

You are right, we removed the relation to precipitation and flooding to keep it more focused on heat.

*Line 97: "using **data from** a densely-distributed"*

We added is correspondingly.

*Line 104: "air and surface temperature**s**"*

We changed it correspondingly.

*Line 110: Add citation for LCZ.*

We added the corresponding citation to reference it.

*Line 115: 1 m*

We changed it correspondingly.

*Line 116: Add citation for ENVI-met model.*

We added the citation for ENVI-met there: "Bruse et al, 2022".

*Line 116: You cite a conference abstract (Eingrüber et al., 2022b) without any information on how the 3D data of your model domain was obtained.*

We have removed the citation from the conference abstract now, and added a citation with all the information on the 3D data enrichment for the model domain setup based on field measurements, available cadastral databases and remote sensing.

*Line 119: Campbell? Please provide more information on the measurements, its instruments etc. Or provide a citation where everything can be found.*

Campbell Scientific is the manufacturer of the research-grade meteorological station that we installed in the urban park. We added an appropriate citation now to a paper where all the relevant information on all the measurement variables, measurement heights, site characteristics of the station etc. are provided in more detail. We also adapted the sentence to "[…] station of the manufacturer Campbell Scientific that we installed in the urban park to […]" to make it clearer.

*Lines 120-123: What are the results of the validation (Eingrüber et al., 2022b)? A conference paper is cited without any results. I think this sentence/citation is not necessary, as in the next sentence a Journal article is cited and information about the accuracy is provided.*

We have removed the citation of the conference paper now. You are right that is not necessary, as all relevant information can be found in the journal article cited in the next sentence, where also the value of the accuracy index NSE is given.

*Lines 120-124: Model evaluation of what? Air temperature?*

Yes, of air temperature. We added it correspondingly.

*Line 122: I think in Eingrüber et al., 2023b the NSE is given with 0.91?*

Thanks, we corrected it correspondingly.

*Line 125: °C*

We corrected it correspondingly.

*Line 126: I did not find any information about this Gumbel distributions in the conference abstract of Eingrüber et al. 2023a. Is this of relevance for the ENVI-met modelling? Or only for the 20-year heat extreme? I don't understand why this citation is added as there is no relation?*

This Gumbel extreme value distribution is not relevant for ENVI-met modelling itself, but was used to determine the study period (the 20-year heat event) which was simulated with the model. We changed the citation now to a paper where the method of deriving the recurrence intervals

(frequency-magnitude relationship) according to the Gumbel extreme value distribution is presented, and the corresponding formula given.

*Line 140: add units*

We added the unit for all instances there.

*Line 146: roughness length = Z0 in line 148?*

Yes, we also added $Z_0$ in this line now.

*Line 156: … "(39 % substrate to 61 % concrete) **and** is used" ?*

We added it correspondingly.

2. Methods:

*Line 174: Sometimes you have a space between Number and %, sometimes not.*

We have now made it uniform for all instances in the paper.

*Line 175: New paragraph*

We changed it correspondingly.

*Line 176: Model, Company?*

Vernier is the company name. The details on the technical specifications of the model PYR-BTA are described in the given web reference. We added the word "from the company" to make it clearer (https://www.vernier.com/product/pyranometer/).

*Line 179: "… **on** a day with clear-sky…"*

We changed it correspondingly.

*Lines 184-192: New paragraph and maybe re-arrange entire section with soil characteristics after hydraulic conductivity at line 175.*

Thank you very much for the good idea. We inserted a new paragraph now for that section, as well as re-arranged the section with the soil characteristics after the section of the soil hydraulic conductivity and soil moisture (and before the radiation/albedo section) so that all soil parameters are now described in a line (and not interrupted by radiation anymore).

*Line 188: "As description of the extent to which the surface deviates from a completely flat surface by elevations…" Which surface? The GPP surface?*

Yes, correct, the GGP surface is meant. We added "GGP surface" to the sentence now.

*Line 190/191: How is the profile meter defined? Does the orientation of the profile meter play a role? Or is it simply an approximation?*

As the geometrical structure of the GGPs is a quadrilateral, the evenly repeating parallelograms are characterized by full point symmetry in itself, where the center of symmetry is the intersection point of the diagonals. Thus, the orientation of the profile meter does not play a role in a geometrical sense. If you rotate the profile meter or if you shift up or down the profile meter by a few centimeters, the number of substrate/concrete edges stays the same. We have added a short explanation on this in a new sentence now: "As the geometrical structure of the GGPs are evenly repeating parallelograms characterized by full point symmetry in itself, the orientation of the profile meter does not play a role for the number of substrate/concrete edges."

*Line 196: "parameters" instead of "parameterizations" (Compare Line 194)*

With this sentence, we mean that the three individual soil material parameterizations of GGPs, sand and gravel were combined to a vertical profile (while the sentence before was focusing on the parameters of these three parameterizations). Thus, we kept the wording as it is, but adapted the formulation to make it easier to understand this relationship now: "These three individually developed soil and surface material parameterizations for GGPs, sand and gravel were then combined to a vertical soil profile […]".

*Line 201: Why emissivity of 0.9? References?*

We changed the sentence and added the corresponding reference now: "[…], and an emissivity of 0.9 was assumed according to the DBManager and in agreement with PELUSO et al. (2022).".

*Lines 226-228: "While GGPs were set for all side streets, the lane widths of the main traffic axes Volksgartenstraße (double-avenue in the middle of the street) and Vorgebirgsstraße **were** measured to determine the number of sealed grid cells in the model domain."*

We have changed it correspondingly.

*Line 244: References for Python and Leonardo DataStudio.*

We have now added the corresponding reference as well as provided the version number 3.9.

*Line 246: Reference for R.*

We have now added the reference for R as well as provided the corresponding version number R-4.3.0.

*Lines 252-254: I don't think you have to describe how you created all results such as how you calculated the difference of mean temperature.*

I fully agree. We have removed that sentence now.

*Line 253: "The hourly layers  were averaged using the raster calculator for individual days and the entire simulation period." What does "hourly layers per day" mean?*

We have removed the expression "per day" which makes no sense, and slightly rephrased the sentence to make the temporal aspect more precise.

*Lines 254-256: Rephrase.*

We have rephrased the sentence by splitting it and restructuring it: "To analyse, if GGPs not only show a cooling effect on air temperature but also lead to an increase in thermal outdoor comfort, biometeorological indices are calculated. While a decrease in temperature increases thermal outdoor comfort, an increase of relative humidity due to GGP evapotranspiration as well as an increase in reflected, secondary radiation due to the higher albedo of GGPs might reduce thermal comfort. To quantify the overall effects of GGPs on comfort of human organisms, […]".

*Lines 257-258: References for PET, UTCI, and BIO-met.*

We have added the relevant references for PET, UTCI and BIO-met now: "BRÖDE et al, 2011; HÖPPE, 1999; BRUSE et al, 2022".

*Line 259-261: Not necessary.*

You are right, this is redundant. We removed the sentence now.

3. Results:

*Line 264: "surface temperature**s from** -2.00 K up to -8.26 K…"*

We have changed it correspondingly.

*Line 268: Is this now the comparison of all grid cells or only the grid cells with GPPs? (Compare with line 244).*

Yes, this is now the comparison of all surface grid cells in the entire model domain. We have slightly changed the wording to make it clearer: "On average for all surface grid cells of the entire 16 ha study area, […]".

*Lines 264-…: Please always define if it is air or surface temperature (Ta & Ts), not only "temperature".*

We have now changed "temperature" to either Ta or Ts for all instances. Also, we abbreviated Ta and Ts for all instances from (old) line 264 onwards now.

*Line 277: "…(19th July 2022), the surface temperature was…" not sure about, but do you mean Ta?*

Very good attentive observation, of course we mean $T_a$. We have changed it now.

*Line 284: Move to discussion.*

We have moved this to the discussion section now, and therefore slightly rephrased the sentence to fit into the discussion context.

*Line 288: "."*

We removed this now.

*Line 295: "S2 and S3 also show a smaller IQR of 9.27 °C and 9.47 °C**, respectively**."*

We have changed it correspondingly.

*Line 297: Maybe add a table with the statistical results (t-, and p-value) to the Appendix.*

We have now added a corresponding table to the Appendix (Appendix 2) with all t-test results of the paper for all intercomparisons for Ta and Ts like shaded-unshaded, daytime-nighttime, etc.

*Section 3.1: Please use paragraphs for Ts and Ta.*

Good suggestion, we have split this section into three paragraphs now: One for Ts, one for Ta, and for the comparison of both.

*Line 303: "(4.09 K more for S3)". More?*

We have changed the wording so that the relation of the cooling effects is described in a more precise way now: "With -4.4 K on average, daytime cooling effects are much more pronounced than at nighttime for S2 (-4.09 K for S3).".

*Line 304: " Since p-values are lower during the day than at night  cooling effects are stronger and more significant during the day."*

We have rephrased the sentence as you suggested. But the information that the cooling effects are still significant also during nighttime was cut out. Therefore, we added this formulation: "[…] during the day, but still significant for nighttime.".

*Line 315: Again, all grid cells? Or only the GPP grid cells as written in the figure caption?*

This is only for the atmospheric grid cells above GGP pixels. We adjusted it accordingly now: "[…] based on the 72 hourly values calculated from the average of all atmosphere grid cells above GGP pixels (for 1 m, 3 m and 5 m height above ground level) […]". We also changed it accordingly in the Ts section so that it becomes clear for both instances.

*Line 320: 72 values or timesteps?*

Both, we have 72 values and 72 timesteps. One value for each hourly output interval of the simulation for the 3-day period (72 hours). These 72 values were used to calculate the temporal average. The 72 values in turn were calculated from the average of all atmosphere grid cells above GGP pixels (for 1 m, 3 m and 5 m height above ground level) for each hour. We have adjusted the sentence to reduce confusion with this: "On temporal average of the 72 hourly mean values of all atmosphere grid cells 1 m above GGP pixels, Ta is 1.08 K cooler than in S1.". We also changed it exactly in the same way in the Ts section so that it becomes clear for both instances.

*Line 330: New paragraph or even section? In general, why summarize results? Isn't this a part for conclusion?*

You are right, it is no necessary to summarize the results here. As this summary with the same information is already given in the conclusions section, we have deleted this redundancy here now. We also separated the results from the biometeorological indices in a paragraph afterwards.

*Line 330-335: "ensible  and  soil heat flux**es**  **decreased** due to  GGP implementation, while  sensible heat flux, relative humidity and soil water content  decreased due to the unsealing,  increased LAD and  different material properties of the GGPs."*

Thanks for the suggestion which we have taken over and slightly added/adjusted some words to make the causality even more precise.

*Line 335-337: Rephrase.*

We rephrased this sentence now also using the abbreviation for Ta: "To analyse, if the cooling effect of GGPs on Ta also leads to an increase in thermal outdoor comfort despite relative humidity and reflected, secondary radiation increases, thermal comfort indices have been calculated.".

*Line 339: Does thermal comfort improve in the entire study area (Fig. 16?). It is a bit difficult to distinguish between the different classes in Figure 16. For me, it appears that a lot of pixels have a value between -0.31 K and 0.33 K, which does not seem to be the entire study area.*

Thermal comfort does not improve in the entire study area. Especially in the urban park area and in some inner courtyards of building blocks, no significant changes in UTCI can be observed. Our causality focusses on the areas where GGPs have been implemented. Therefore, we adjusted the sentence to make this clearer: "It becomes clear that thermal comfort is significantly increased on areas where GGPs have been implemented. Especially in the street canyons, a decrease of UTCI by up to -2.6 K was calculated. On spatial average for all grid cells in 1 m height above GGP pixels, an UTCI improvement of -1.7 K was found.".

*Line 340: When and how often did the UTCI classes changed from very strong heat stress to strong heat stress? Pixel-wise or on spatial average? For individual hours or on temporal average?*

This expression is on temporal and spatial average. We modified the sentences accordingly to: "On temporal and spatial average for all grid cells in 1 m height above GGP pixels, an UTCI improvement of -1.7 K was found. In S2, the perceived temperatures drop from very strong heat stress to only strong heat stress on temporal and spatial average for all grid cells in 1 m height above GGP pixels. Also, the Physiological Equivalent Temperature (PET) shows a decrease of the perceived temperature

on temporal and spatial average of all areas where GGPs have been implemented by -4.6 K and up to -6.8 K.".

*Lines 343-353: Move to Conclusion?*

As the statistical significance tests are a method applied in this paper and were described in the method section, we here present the results of this method, which is the finding that the H0 hypotheses can be rejected and the alternative hypotheses can be accepted. Therefore, we think that these findings are placed correctly here in the results chapter synchronic to the representation on the methods chapter. We slightly adjusted the formulations of this paragraph so that it is not formulated like conclusions. Of course, we highlight these statements in the conclusions later.

4. Discussion:

*Lines 360-364: Surface or air temperature? Not clear, as first it is written "At the surface level", but then it is compared to Ta.*

We mean Ta in 1 m height above ground level. We changed it accordingly to: "For Ta in 1 m height above ground level, […]".

*Line 383: Again, which differences are higher during the day? I think its Ts, right?*

Yes, Ts is correct. We modified it accordingly to: "The observed significant differences in the Ts cooling effects […]".

*Line 387: "The surface temperatures  **of** the grass areas"*

We changed it to: "Ts of grass areas also […]".

*Lines 384-388: For my understanding: In your study the sensible heat flux was reduced by 130 Wm-2, which means from 140 Wm-2 to 10 Wm-2, and in TAKEBAYASHI & MORIYAMA (2009) it was reduced from 250 Wm-2 by 100 Wm-2 to 150 Wm-2. Is this correct? But isn't there a big difference between 10 Wm-2 and 130 Wm-2? Is there an overview of all fluxes?*

Sorry that our wording was a bit confusing regarding the expression of the change in TAKEBAYASHI & MORIYAMA (2009). It was not reduced from 250 W/m² by 100 W/m² to 150 W/m². The reduction itself ranged between 100 W/m² and 150 W/m². We changed it and have split the sentence to clarify the wording: "[…] a reduction of sensible heat flux ranging between 100 W/m² and 150 W/m². This change in sensible heat flux agrees well with the magnitude of the change in sensible heat flux of our study […]".

*Lines 399-401: If I understood correctly, you wrote about Ts in the adjacent sentences. Here you mention thermal indices, which are less affected by Ts but rather by Ta. So, I don't get the context. Or is this already part of the following sentences, where you start writing about Ta?*

Yes, in the sentences before, we talked about Ts. You are absolutely right that this order is confusing as the thermal comfort indices are most affected by Ta and should therefore better be placed after the Ta paragraph. Therefore, we moved the sentences on thermal indices to the end of the Ta section now, and also separated the Ts and Ta paragraphs by a line break to make the order more concise.

*Line 404-405: Rephrase: "At nighttime, cooling effects of GGPs on surface temperature are much lower, but significantly higher differences in air temperature occur in both S2 and S3 than during daytime."*

We rephrased it to: "At nighttime, cooling effects of GGPs on Ts are significantly smaller than during daytime, while more significant cooling effects on Ta occur during nighttime in both S2 and S3.".

*Line 409: Ta increase in percentage? I think this is rather uncommon. It does not seem that you use the Kelvin scale? To my understanding, everything else will give arbitrary results, as Celsius has no "absolute zero" point in 0°C. So, its maybe better to avoid using percentage for temperatures changes.*

You are absolutely right, but this formulation should not describe the Ta decrease (which of course should always be expressed in Celsius/Kelvin); however, it should describe the change in the delta Ta. During nighttime, delta Ta ranges from -1.12 K for S2 to -1.00 K for S3. During daytime, it ranges from -1.00 K for S2 to -0.84 K for S3. Delta Ta is therefore 11 to 16 % higher during nighttime compared to daytime. We adapted the formulation so that it gets clear now: "This can explain why delta Ta in S2 and S3 was on average around 11 % to 16 % smaller during daytime compared to nighttime.".

**Tables and Figures:**

*Table 1: "Thermal conductivity [W/(mk)]"*

We corrected it to: "[W/(mk)]".

*Figures 9, 11, and 12: Merge to one figure with subplots a, b, and c? Would make understanding easier. Then you would need only one legend with S1, S2, and S3. In addition, a boxplot of all Ta and Ts differences would be nice to see, not averaged by timestep, just 72x400x400 values to get an impression of the entire "cooling" distribution.*

We have now merged these three figures to one with subplots a, b, and c and one single legend. We have tried to design a boxplot with all Ta and Ts differences (72 x 400 x 400 values) and decided that it is not suitable for the visual identification of differences between the scenarios due to the very high variability of the values. The very long whiskas (and outliers) of the boxplots and the very small boxes itself would then make a visual comprehension very difficult and reduce the statistical interpretability. Therefore, we decided to keep the representation as it is, clearly divided into the boxplots for depiction of the temporal distribution of the mean spatial differences (72 values), and the maps for depiction of the spatial distribution of the mean temporal differences (400 x 400 values).

*Same for figures 13 and 15.*

We have now merged these two figures to one with subplots a and b and one single legend.

*Can this merged figure also be done for UTCI?*

There is only one Figure (difference map) for UTCI in this paper. So, we do not see a figure with which we can merge it. We also do not want to add additional figures for UTCI or PET as the focus of the research questions and hypotheses of this paper is on thermal effects of the GGPs and not directly on biometeorological effects. Thus, we only want to show that one figure of UTCI as some kind of outlook and not present all statistical analyses and comparisons for heights, day/nighttime, etc. also for UTCI.

*Why is figure 10 so small compared to figure 14? You could also merge them*

The size differences were just caused by using the maximum page width to best identify the differences between the curves. We have now better balanced the figure size and also merged the two figures to one with subplots a and b.

*Figure 8 caption: "The color scales correspond to those in Figure 6 and Figure 7." Is it possible to add these color scales to the figure 8? Makes it easier to analyze.*

We have added the two color scales to figure 8 now for an easier understanding.

*Figures 6-8: I wonder about the edges of almost all plots. Also, in comparison to Figure 5. It somehow looks like the model domains have different edges between S1 and S2/S3 (Figure 5)? Does this lead to the high differences at the edges from Figure 6?*

No, the edges of all scenarios S1, S2 and S3 are exactly the same as they are based on the same model domain with the same dimensions, geometries and spatial characteristics. The differences at the edges are typical border effects in ENVI-met which are caused by the empty/nesting border grid

cells. Especially at the borders where no obstacles like buildings are located, higher wind speeds and higher radiation can typically be found than somewhere within the domain. Higher wind speeds can increase the evapotranspiration performance of the GGPs. More radiation (due to no shadings) can also increase the cooling effects of the GGPs as we learned in the paper for the direct intercomparison of shaded and unshaded GGP areas. If micrometeorological conditions are different at the edges, also the energetic effects and cooling potentials of the GGPs differ.

*Figure 16: What about the mean?*

We have added the mean value of -1.13 K to the figure now.

**RC #3:**

The manuscript titled "Simulation of the heat mitigation potential of unsealing measures in cities by parameterizing grass grid pavers for urban microclimate modelling with ENVI-met (V5)" presents an innovative approach to urban heat mitigation through the application of grass grid pavers (GGPs). In light of the increasing frequency of summer heat events and the growing urban population, the study highlights the need for climate adaptation measures that do not require additional space or alter the fundamental function of urban areas. The authors have developed a new GGP parameterization for the ENVI-met V5 microclimate model, filling a research gap in the field. The study includes scenario analyses that assess the cooling potential of GGPs in a high-density urban area in Cologne, Germany, under extreme heat conditions. The realistic implementation scenario also demonstrates substantial cooling effects, confirming that partial unsealing through GGPs can effectively mitigate urban heat, provided that water availability is sufficient.

General Opinion:

The study presented in this manuscript is timely and relevant, particularly as cities worldwide face the challenges posed by climate change and increased urban heat. The use of GGPs as a measure to reduce urban temperatures through unsealing strategies is a contribution to the field of urban climate adaptation. The development of a new parameterization for GGPs within the ENVI-met model is also an advancement. However, there are several issues that need to be addressed to enhance the manuscript's impact and suitability for publication in a journal like Geoscientific Model Development (GMD).

*1. Insufficient Development Contribution: While the development of the GGP parameterization for ENVI-met is a key aspect of the study, it is not sufficiently highlighted. The contribution of this development to the broader field of urban microclimate modelling is underemphasized, and the manuscript could benefit from a more detailed explanation of the parameterization process, including its novelty and significance compared to existing approaches.*

Thank you very much for your valuable feedback. We deal with the novelty and significance of this approach compared to existing approaches in more detail in the second comment and in some of the comments of Reviewer 2 above. To better highlight the contribution of this development, we have now better emphasized in several parts of the manuscript (Abstract, Introduction, Discussion, Conclusion) to better point out the novelty of the parameterization based on in-situ measurements and to show that this parameterization could also be used in the broader field of urban microclimate modelling: "To fill this research gap, we here present a new GGP model parameterization developed for the fluid dynamics microclimate ENVI-met model based on field measurements with double-ring infiltrometers etc. which can also be implemented in other microscale models in the field of urban climatology.", "The results of this study prove the suitability of the newly-developed parameterization of GGPs for microclimate modelling in ENVI-met to fill the identified research gap which could also be used in the broader field of urban microclimate modelling and implemented in models on a similar scale like PALM-4U or MITRAS.", "As GGPs have never been parameterized for microclimate modelling with ENVI-met before, a new parameterization was developed using in-situ measurements to fill this research gap which can also be implemented in other urban microclimate models.".

*2. Lack of Review on Similar Parameterizations: The manuscript does not include a review of similar parameterizations in other microscale urban climate models, such as PALM, MITRAS, or ENVI-met itself. A thorough comparison with these models would provide context for the significance of the new GGP parameterization and could strengthen the paper's contribution to the field.*

For ENVI-met, we review all available parameterizations: "However, GGPs have not been parameterized yet for microclimate modelling with the established numerical ENVI-met model. Until now, microclimate modelling studies only represented GGPs or similar surfaces as a separate mixture of pure grass and pure concrete in a stripe or chess board arrangement […] [PELUSO et al., 2022; HOFFMANN & GEISSLER, 2022].", "TEOH et al. (2022) used the surface layer for grass already parameterized in ENVI-met for the GGPs. REZK (2021) adjusted the albedo and root depth of the pre-parameterized grass. In the study conducted by BATTISTI et al. (2018), alternating grass and concrete strips were implemented in the model domain to roughly approximate the characteristics of GGPs […] (SAITO et al., 2015).", "Modelling of GGPs in ENVI-met has only been carried out sporadically by a conceptual implementation separate grass and pavement arrangements, but GGPs have never been parameterized in any study [JIA & WANG, 2021; BATTISTA et al., 2022]. Thus, microclimate modelling of parameterized GGPs in ENVI-met to analyze cooling effects and adaptation potentials in dense urban environments represents a novelty. To fill this research gap, this study presents a new parametrization of GGPs based upon in-situ measurements. […]".
For the METRAS model, we added a citation now to a study where GGPs were parameterized on a different spatial scale: "In a modelling analysis by BÖTTCHER (2017) using the METRAS model for the city of Hamburg, interlocking pavers with a grass component have been parameterized and simulated to assess the climatic impacts for the region. The parameterization was conducted for an intra-urban and not obstacle-resolving mesoscale and found a slight cooling effect of 1 to 2 K on the surface and less than 1 K in the air.".

*3. Need for Discussion on Cooling Effect Measures: The manuscript could be improved by discussing the differences in cooling effect measures, specifically air temperature (Tair), Universal Thermal Climate Index (UTCI), and surface temperature (Tsurface). This discussion would benefit from referencing studies such as https://doi.org/10.1175/BAMS-D-20-0193.1 and https://doi.org/10.1016/j.enbuild.2023.113324, which address the complexities of these different metrics in the context of urban heat mitigation.*

We have added a corresponding paragraph in the discussion section now to deal with the complexity of different metrics to describe cooling effects, and therefore also used your suggested literature: "The identification and expression of cooling effects in the context of urban heat mitigation is complex, and largely depends on the crucial selection of thermal metrics which are not generalizable (MIDDEL et al., 2021). While metrics like Ts can directly represent the physical processes at energy conversion surfaces resulting in the greatest effects, Ta as an integrative energy description of a volume is less sensitive due to external effects like wind flow, but more relevant for pedestrians and better describes the overall cooling effects for an urban environment. Mean Radiant Temperature and thermal comfort indices like UTCI or PET are well suited for describing heat perception and stress of human individuals and take into account parameters like clothing and metabolism parameters as well as personal characteristics like body mass index. Nevertheless, comfort indices have limitations with regard to generalization and transferability for people with different ages, sizes, gender and weight, and therefore are subjective to assumed standardizations. Quantification of cooling effects should therefore always use different metrics to instead of single ones to describe the direct physical causalities as well as the integrative effects and the consequences for perception (ANDERS et al., 2023).".

*4. Reiterating Points from Reviewer Comments: Most of the issues I recognized have already been listed by RC2 (https://egusphere.copernicus.org/preprints/2024/egusphere-2024-697/egusphere-2024-697-RC2-supplement.pdf). I agree with the points raised, particularly regarding the need for a more in-depth discussion of the modelling approach and its implications for the accuracy and applicability of the results.*

All the changes based on the RC2 can be found above, also regarding the discussion of the approaches.

---

## Author Response (AR2)

**Response Technical Corrections:**

**Editor Comment:**

*Although the reviewers have raised two technical corrections, I believe that the paper is ready for publication after correcting these two issues:*

*1- Consider the quality standards of the figures:*
*Review file validation: With the next file upload request, please rename the material in the uploaded supplement \*.zip to e.g., "Figure S1" and "Figure S2" (please see: https://www.geoscientific-model-development.net/submission.html#assets -> section "Supplements").*

*2- Update the citations of the microscale models:*
*RC1: I believe that authors worked on the issues I have raised in my first review report. The manuscript is now much better than the first version. There is only one minor issue is that authors mentioned the models without proper citations. For example, the model PALM-4U and MITRAS were mentioned without citation. I guess the proper citations are also GMD articles: https://doi.org/10.5194/gmd-13-1335-2020 and https://doi.org/10.5194/gmd-11-3427-2018*
*RC2: Accepted as is*

*I would like to thank the three reviewers who carefully revised the paper and contributed to the improvement of the paper. I Also would like to thank the authors for the productive discussions and improvements they did for the manuscript.*

1.) We have renamed the figures in the supplementary material to "Figure S1" and "Figure S2" now according to the standards. We have therefore reuploaded the supplementary ZIP file and adjusted the corresponding citations of the two supplementary figures in the manuscript text now.

2.) We have now added the corresponding citations for the models PALM-4U and MITRAS as suggested:
Maronga, B., Banzhaf, S., Burmeister, C., Esch, T., Forkel, R., Fröhlich, D., Fuka, V., Gehrke, K. F., Geletič, J., Giersch, S., Gronemeier, T., Groß, G., Heldens, W., Hellsten, A., Hoffmann, F., Inagaki, A., Kadasch, E., Kanani-Sühring, F., Ketelsen, K., Khan, B. A., Knigge, C., Knoop, H., Krč, P., Kurppa, M., Maamari, H., Matzarakis, A., Mauder, M., Pallasch, M., Pavlik, D., Pfafferott, J., Resler, J., Rissmann, S., Russo, E., Salim, M., Schrempf, M., Schwenkel, J., Seckmeyer, G., Schubert, S., Sühring, M., von Tils, R., Vollmer, L., Ward, S., Witha, B., Wurps, H., Zeidler, J., and Raasch, S.: Overview of the PALM model system 6.0, Geosci. Model Dev., 13, 1335–1372, https://doi.org/10.5194/gmd-13-1335-2020, 2020.
Salim, M. H., Schlünzen, K. H., Grawe, D., Boettcher, M., Gierisch, A. M. U., and Fock, B. H.: The microscale obstacle-resolving meteorological model MITRAS v2.0: model theory, Geosci. Model Dev., 11, 3427–3445, https://doi.org/10.5194/gmd-11-3427-2018, 2018.

We would like to thank the reviewers for their valuable feedback and suggestions improving the manuscript, as well as the editor for the professional implementation and coordination of the peer-review process and the thorough evaluation of the reviews and feedback.